# Synthio: Augmenting Small-Scale Audio Classification Datasets with Synthetic Data

**Sreyan Ghosh**[♦♣*], **Sonal Kumar**[♣*], **Zhifeng Kong**[♦], **Rafael Valle**[♦], **Bryan Catanzaro**[♦]
**Dinesh Manocha**[♣]
[♦]NVIDIA, CA, USA, [♣]University of Maryland, College Park, USA
{sreyang, sonalkum}@umd.edu [*]equal technical contribution.
Project: `https://sreyan88.github.io/Synthio/`

## Abstract

We present **Synthio**, a novel approach for augmenting small-scale audio[1] classification datasets with synthetic data. Our goal is to improve audio classification accuracy with limited labeled data. Traditional data augmentation methods, which introduce synthetic perturbations (e.g., injecting noise or masking segments), fail to replicate the inherent variability found in real-world audio. To address this shortcoming, we propose to augment the dataset with synthetic audio generated from text-to-audio (T2A) diffusion models. However, synthesizing effective augmentations is challenging because not only should the generated data be *acoustically consistent* with the underlying small-scale dataset, but they should also have sufficient *compositional diversity*. To overcome the first challenge, we align the generations of the T2A model with the small-scale dataset using preference optimization. This ensures that the acoustic characteristics of the generated data remain consistent with the small-scale dataset. To address the second challenge, we propose a novel caption generation technique that leverages the reasoning capabilities of Large Language Models to (1) generate diverse and meaningful audio captions and (2) iteratively refine their quality. The generated captions are then used to prompt the aligned T2A model. We extensively evaluate Synthio on ten datasets and four simulated limited-data settings. Results indicate our method consistently outperforms all baselines by 0.1%-39% using a T2A model trained only on weakly-captioned AudioSet.

## 1 Introduction

Audio classification is the foundational audio processing task of understanding the input audio and assigning it to one or multiple predefined labels. However, training audio classification models requires a lot of high-quality labeled data, which is not always readily available (Ghosh et al., 2022). Manually collecting and annotating large-scale audio datasets is an expensive, time-consuming, and noisy process (Nguyen et al., 2017; Martín-Morató & Mesaros, 2021), and recent concerns about data privacy and usage rights further hinder this process (Ren et al., 2023).

Data augmentation, which involves expanding original small-scale datasets with additional data, is a promising solution to address data scarcity. Traditional augmentation techniques attempt to diversify audio samples by applying randomly parameterized artificial transformations to existing audio. These methods include spectral masking (Park et al., 2019), temporal jittering (Nanni et al., 2020), cropping (Niizumi et al., 2021), mixing (Seth et al., 2023; Ghosh et al., 2023b; Niizumi et al., 2021) and other techniques (Al-Tahan & Mohsenzadeh, 2021; Saeed et al., 2021; Manocha et al., 2021; Cherep & Singh, 2025). Although these methods have been effective, they focus on surface-level patterns in the data rather than capturing the fundamental mechanisms that drive real-world data generation (Cherep & Singh, 2025). As a result, they statistically modify the data without directly influencing the causal mechanisms that produced it, leading to high correlations between augmented samples and limited control over diversity.

Generating synthetic data from pre-trained text-to-audio (T2A) models addresses the limitations of standard data augmentation techniques while retaining their strengths of *universality*, *controllability*, and *performance* (Trabucco et al., 2024). The recent success of generative models makes

---

[1]We use "audio" to refer to acoustic events comprising non-verbal speech, non-speech sounds, and music.

this approach particularly appealing (Long et al., 2024; Evans et al., 2024b). However, generating synthetic audio presents unique challenges due to the complexity of waveforms and temporal dependencies (Ghosh et al., 2024b). We highlight the 3 main challenges in generating effective synthetic data for audio classification: **i) Consistency with the original data:** Synthetic audio that does not align acoustically with the original dataset can hinder effective augmentation and may cause catastrophic forgetting (Geiping et al., 2022). This misalignment includes spectral, harmonic, and other inherent acoustic characteristics not easily controlled through prompts. Maintaining consistency with T2A models trained on internet-scale data remains a challenge, and standard fine-tuning can often lead to overfitting (Weili et al., 2024). **ii) Diversity of generated data:** Ensuring compositional diversity in the generated synthetic data (e.g., sound events, temporal relationships, background elements, etc.) is critical for effective augmentation. Additionally, a lack of diversity can lead to poor generalization and learning of spurious correlations, impacting performance. Simple, hand-crafted prompts (e.g., "Sound of a metro") often result in repetitive patterns, and creating diverse, meaningful prompts is labor-intensive. Complex prompts can generate audios that do not preserve the original label. **iii) Limitations of current T2A models:** T2A models often struggle to generate diverse audios and follow details in prompts. This is largely due to the lack of large-scale, open-source datasets for training, as well as the inherent complexity of non-speech audio domains (Ghosal et al., 2023). These limitations highlight the need for more advanced approaches for synthetic data generation in audio.

**Our Contributions.** To address these challenges, we propose **Synthio**, a novel, controllable and scalable approach for augmenting small-scale audio classification datasets with synthetic data.

Our proposed approach has 2 main steps: i) *Aligning the Text-to-Audio Models with Preference Optimization:* To generate synthetic audios with acoustic characteristics consistent with the small-scale dataset, we introduce the concept of ***aligning teaching with learning preferences***. Specifically, we align the generations of the T2A model (acting as the teacher) with the target characteristics of the small-scale dataset using preference optimization. This approach ensures that the synthetic audios reflect the acoustic properties of (or *sound similar to*) the downstream dataset, enabling the classification model (the student) to perform well on test data with similar characteristics. To achieve this, we train a diffusion-based T2A model with preference optimization, where audios generated from Gaussian noise are treated as losers and audios from the downstream dataset are treated as winners.

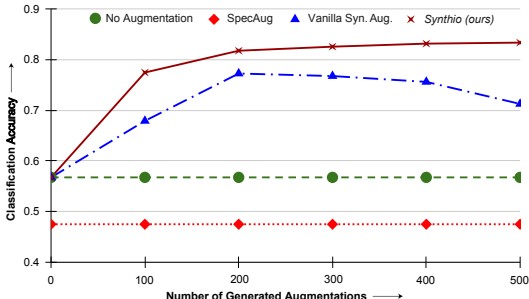

Figure 1: Performance comparison of Synthio with other augmentation methods on down-sampled ESC-50 (100 samples). Traditional augmentation, such as SpecAug, degrades performance on small-scale datasets. Naive synthetic augmentation outperforms traditional methods significantly but plateaus with higher sample counts. Synthio further enhances performance by generating consistent and diverse synthetic data.

ii) *Generating Diverse Synthetic Augmentations:* To generate diverse audios for augmentation, we introduce the concept of ***language-guided audio imagination*** and imagine novel acoustic scenes with language guidance. Specifically, we generate diverse audio captions that are then used to prompt T2A models to generate audios with varied compositions. To achieve this, we propose *MixCap*, where we prompt LLMs iteratively to generate captions combining existing and new acoustic components. Additionally, we employ a *self-reflection module* that filters generated captions and prompts the LLM to revise those that do not align with the intended label. To summarize, our main contributions are:

1. We introduce **Synthio**, a novel data augmentation approach for audio classification that expands small-scale datasets with synthetic data.
2. We evaluate Synthio across 10 datasets in 4 simulated low-resource settings. Synthio outperforms all baselines by 0.1%-39%.
3. We conduct an in-depth analysis of the generated augmentations, highlighting Synthio's ability to produce diverse and consistent data, its scalability, and its strong performance on complex tasks such as audio captioning.

## 2 RELATED WORK

**Data Augmentation for Audio and Beyond.** Expanding or augmenting small-scale datasets with additional data has been widely studied in the literature. Traditional augmentation methods, which

apply randomly parameterized artificial transformations to data during training, remain the most common approach across language Wei & Zou (2019); Karimi et al. (2021), vision (Shorten & Khoshgoftaar, 2019; Wang et al., 2017; Yun et al., 2019), and audio (Park et al., 2019; Spijkervet, 2021). For audio, specific techniques include SpecAugment, adding background noise, reverberation, and random spectrogram transformations. With the emergence of generative models, synthetic data augmentation has been increasingly adopted for language (Ghosh et al., 2023a; 2024c; Chen et al., 2021) and vision (Trabucco et al., 2024; Zhao et al., 2024), proving to be more effective than traditional methods. These approaches generally incorporate explicit steps to ensure the consistency and diversity of generated augmentations. In contrast, the application of synthetic data to audio and speech remains underexplored. Recent attempts include generating synthetic captions for improving audio-language pre-training (Xu et al., 2023), improving T2A models with synthetic captions (Kong et al., 2024) and scene classification (Ronchini et al., 2024; Feng et al., 2024; Cherep & Singh, 2025).

**Few- and Zero-Shot Audio Classification.** Few-shot audio classification focuses on training models to classify audio samples with very limited labeled data per class, often leveraging transfer learning or meta-learning approaches (Zhang et al., 2019; Wang et al., 2021; Heggan et al., 2022). In contrast, zero-shot audio classification enables models to generalize to unseen categories without direct training on those classes, relying on learned representations or external knowledge (Xie & Virtanen, 2021; Elizalde et al., 2023). Synthetic data research complements these by generating additional labeled data, improving model performance under low-resource settings while addressing data scarcity without directly requiring labeled instances from the target categories.

**Text-to-Audio Generation.** In recent years, there has been a significant surge in research on text-to-audio (T2A) models. The most popular architectures include auto-regressive models based on codecs (Kreuk et al., 2023; Copet et al., 2024) and diffusion models Liu et al. (2023); Ghosal et al. (2023); Evans et al. (2024a). Clotho (Drossos et al., 2020) and AudioCaps (Kim et al., 2019) remain the largest human-annotated datasets for training these models. However, large-scale datasets for T2A model training are still scarce. Recently, Yuan et al. (2024) synthetically captioned AudioSet (Gemmeke et al., 2017), demonstrating its effectiveness for training T2A models. For downstream adaptation, earlier works have primarily relied on Empirical Risk Minimization (ERM). Majumder et al. (2024) introduced preference optimization for T2A models, creating a synthetic preference dataset based on scores provided by a CLAP model (Elizalde et al., 2023).

## 3 BACKGROUND

**Diffusion Models.** Diffusion models consist of two main processes: a forward process and a reverse process. Given a data point $x_0$ with probability distribution $p(x_0)$, the forward diffusion process gradually adds Gaussian noise to $x_0$ according to a pre-set variance schedule $\gamma_1, \cdots, \gamma_T$ and degrades the structure of the data. We request readers to refer to App. A.1 for more details on diffusion models.

**Reward Modeling.** Estimating human preferences for a particular generation $x_0$ (hereafter treated as a random variable for language), given the context $c$, is challenging because we do not have direct access to a reward model $r(c, x_0)$. In our scenario, we assume only ranked pairs of samples are available, where one sample is considered a "winner" ($x_0^w$) and the other a "loser" ($x_0^l$) under the same conditioning $c$. Based on the Bradley-Terry (BT) model, human preferences can be modeled as:

$$p_{\text{BT}}(x_0^w \succ x_0^l | c) = \sigma(r(c, x_0^w) - r(c, x_0^l)) \tag{1}$$

where $\sigma$ represents the sigmoid function. The reward model $r(c, x_0)$ is parameterized by a neural network $\phi$ and trained through maximum likelihood estimation for binary classification:

$$L_{\text{BT}}(\phi) = -\mathbb{E}_{c, x_0^w, x_0^l}\left[\log \sigma(r_\phi(c, x_0^w) - r_\phi(c, x_0^l))\right] \tag{2}$$

Here, prompt $c$ and data pairs $(x_0^w, x_0^l)$ are drawn from a dataset labeled with human preferences.

**RLHF :** (Christiano et al., 2017) The goal of RLHF is to optimize a conditional distribution $p_\theta(x_0|c)$, where $c \sim \mathcal{D}_c$, such that the latent reward model $r(c, x_0)$ is maximized. This is done while regularizing the distribution through the Kullback-Leibler (KL) divergence from a reference distribution $p_{\text{ref}}$, resulting in the following objective:

$$\max_{p_\theta} \mathbb{E}_{c \sim \mathcal{D}_c, x_0 \sim p_\theta(x_0|c)}[r(c, x_0)] - \beta D_{\text{KL}}[p_\theta(x_0|c) \| p_{\text{ref}}(x_0|c)] \tag{3}$$

Here, the hyperparameter $\beta$ controls the strength of regularization.

**DPO :** DPO directly optimizes the conditional distribution $p_\theta(x_0|c)$ to align data generation with the preferences observed in (any form of) feedback. The goal is to optimize the distribution of generated data such that it maximizes alignment with human preference rankings while maintaining consistency with the underlying reference distribution $p_{\text{ref}}(x_0|c)$.

The optimal solution $p_\theta^*(x_0|c)$ for the DPO objective can be expressed as:

$$p_\theta^*(x_0|c) = p_{\text{ref}}(x_0|c)\frac{\exp(r(c,x_0)/\beta)}{Z(c)} \tag{4}$$

where $Z(c)$ is the partition function, defined as:

$$Z(c) = \sum_{x_0} p_{\text{ref}}(x_0|c)\exp(r(c,x_0)/\beta) \tag{5}$$

This term ensures proper normalization of the distribution, and $\beta$ controls the regularization, balancing between adherence to the reference distribution and preference maximization. The reward function $r(c, x_0)$ is then reparameterized as:

$$r(c,x_0) = \beta \log \frac{p_\theta^*(x_0|c)}{p_{\text{ref}}(x_0|c)} + \beta \log Z(c) \tag{6}$$

Using this reparameterization, the reward objective can be formulated as:

$$L_{\text{DPO}}(\theta) = -\mathbb{E}_{c,x_0^w,x_0^l}\left[\log \sigma\left(\beta \log \frac{p_\theta(x_0^w|c)}{p_{\text{ref}}(x_0^w|c)} - \beta \log \frac{p_\theta(x_0^l|c)}{p_{\text{ref}}(x_0^l|c)}\right)\right] \tag{7}$$

By optimizing this objective, DPO enables direct preference learning, optimizing the conditional distribution $p_\theta(x_0|c)$ in such a way that it better reflects human preferences, as opposed to traditional approaches that optimize the reward function first and then perform reinforcement learning.

**DPO for Diffusion Models:** Very recently, Wallace et al. (2024) propose a formulation for optimizing diffusion models with DPO. The primary issue with optimizing diffusion with DPO is that the distribution $p_\theta(x_0|c)$ is not tractable due to the need to consider all possible diffusion paths leading to $x_0$. To address this, Wallace *et al.* propose to leverage the evidence lower bound (ELBO) to incorporate latents $x_{1:T}$, which represent the diffusion path. The reward $R(c, x_{0:T})$ accounts for the entire sequence, leading to the reward function:

$$r(c,x_0) = \mathbb{E}_{p_\theta(x_{1:T}|x_0,c)}[R(c,x_{0:T})] \tag{8}$$

Instead of directly minimizing the KL-divergence as typically done, they propose to utlize the upper bound of the joint KL-divergence $\mathbb{D}_{KL}[p_\theta(x_{0:T}|c)||p_{\text{ref}}(x_{0:T}|c)]$. This is integrated into the optimization objective, enhancing the practicality of training diffusion models with preferences. The new objective, aiming to maximize the reward and match the distribution of the reverse process of $p_\theta$ to the reference model $p_{\text{ref}}$, is given by:

$$\max_{p_\theta}\mathbb{E}_{c,x_0\sim p_\theta(x_{0:T}|c)}[r(c,x_0)] - \beta\mathbb{D}_{KL}[p_\theta(x_{0:T}|c)||p_{\text{ref}}(x_{0:T}|c)] \tag{9}$$

Training efficiency is improved by approximating the intractable reverse process using a forward approximation $q(x_1:T|x_0)$. The DPO then integrates this into the loss function, which involves comparing the log likelihood ratio of the probabilities under $p_\theta$ and $p_{\text{ref}}$ for winning and losing paths:

$$L_{\text{DPO-Diffusion}}(\theta) = -\mathbb{E}_{(c,x_0^w,x_0^l)\sim\mathcal{D}_{\text{pref}}}\left[\log \sigma\left(\beta T \log \frac{p_\theta(x_{1:T}^w|x_0^w)}{p_{\text{ref}}(x_{1:T}^w|x_0^w)} - \beta T \log \frac{p_\theta(x_{1:T}^l|x_0^l)}{p_{\text{ref}}(x_{1:T}^l|x_0^l)}\right)\right] \tag{10}$$

After applying Jensen's inequality to take advantage of the convexity of $-\log \sigma$, we push the expectation outside, allowing us to simplify the objective. By approximating the denoising process with the forward process, the final form of the loss for DPO in diffusion models becomes:

$$L_{\text{DPO-Diffusion}}(\theta) = -\mathbb{E}_{(c,x_0^w,x_0^l)\sim\mathcal{D}_{\text{pref}},t,\epsilon_t^w,\epsilon_t^l}\left[\log \sigma\left(-\beta T \omega(\lambda_t)\Delta L\right)\right] \tag{11}$$

where $\Delta L$ is the L2 weighted noise estimation losses between the preferred (winner) and less preferred (loser) samples.

## 4 METHODOLOGY

Let $\mathcal{D}_{\text{small}} = \{(a_i, l_i), 1 \le i \le n\}$ be a high-quality, small-scale human-annotated audio classification dataset with $n$ audio-label pairs. Let $\mathcal{D}_{\text{a-c}}$ be a potentially noisy, large-scale weakly-captioned dataset of audio-caption pairs with zero intersection with $\mathcal{D}_{\text{small}}$. Our goal is to train a T2A model $\mathcal{T}^\theta$ using

$\mathcal{D}_{\text{a-c}}$, then use it to generate a synthetic dataset $\mathcal{D}_{\text{syn}}$ and then finally add it to $\mathcal{D}_{\text{small}}$ (now attributed as $\mathcal{D}_{\text{train}}$) to improve audio classification performance. This is accomplished through two key steps: first, aligning the generations from $\mathcal{T}^{\theta}$ with the acoustic characteristics of $\mathcal{D}$small, and second, generating new captions to prompt $\mathcal{T}^{\theta}$ for creating synthetic audio data.

## 4.1 Aligning the Text-to-Audio Model using Preference Optimization

T2A models trained on internet-scale data often generate audio that diverges from the characteristics of small-scale datasets, resulting in distribution shifts. These mismatches can include variations in spectral (e.g., frequency content), perceptual (e.g., pitch, loudness), harmonic, or other acoustic characteristics [2]. This misalignment arises from the non-deterministic nature of T2A generation and it is impractical to provide detailed attributes (like "loud" or "high-pitched") in prompts, as **(i)** there are no scalable methods for extracting specific attributes for each label, and **(ii)** T2A models struggle with accurately following fine-grained prompt details (Wang et al., 2024).

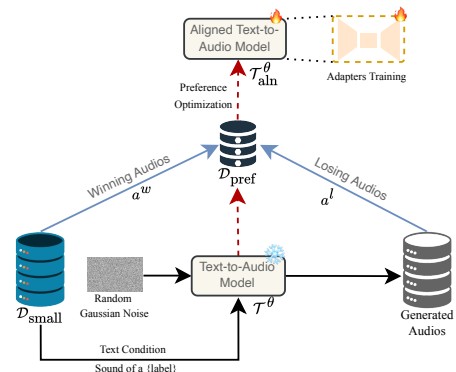

Figure 2: We propose to align the T2A model $\mathcal{T}^{\theta}$ with the small-scale dataset $\mathcal{D}_{\text{small}}$ using DPO. This helps us generate audios with acoustic characteristics aligned to that of $\mathcal{D}_{\text{small}}$.

To address these issues, we propose the concept of ***aligning teaching with learning preferences***. Our approach assumes that the classification model (viewed as the student) performs better when trained on synthetic audio that closely matches the inherent acoustic properties of our high-quality and human-labeled $\mathcal{D}_{\text{small}}$. Thus, we align the generations of the T2A model (viewed as the teacher) to $\mathcal{D}_{\text{small}}$, ensuring that the generated augmentations align with the desired characteristics and *sound similar*, ultimately enhancing the student model's ability to generalize to similarly characterized test data. As shown in Fig. 2, we achieve this using preference optimization (DPO in our case) and align generations of $\mathcal{T}^{\theta}$ with $\mathcal{D}_{\text{small}}$. Unlike standard fine-tuning, which can lead to less diverse outputs and overfitting due to a narrow focus on minimizing loss, preference optimization encourages greater exploration in the model's output space, preventing mode collapse and fostering more diverse augmentations. Additionally, DPO leverages pairwise learning, offering richer training signals compared to the independent outputs used in standard fine-tuning, further mitigating overfitting risks. We detail our two-step approach for DPO optimization below:

**Step 1: Construction of the Preference Dataset.** To create our preference dataset $\mathcal{D}_{\text{pref}} = \{(a_1^w, a_1^l), \cdots, (a_j^w, a_j^l)\}$, we first generate template-based captions for each instance in $\mathcal{D}_{\text{small}}$ in the form: "Sound of a *label*", where *label* is the category associated with the audio. For each instance, we prompt the T2A model $j$ times, with all generations starting from randomly initialized Gaussian noise (generation configuration is detailed in Section 5). Each generated audio is then paired with the corresponding ground-truth audio from the gold dataset. This resulting $\mathcal{D}_{\text{pref}}$ dataset has $n \times j$ instances, where the generated audio is treated as the "loser" and the ground-truth audio as the "winner". This simple approach has proven highly effective in aligning generations by generative models by prior work (Majumder et al., 2024; Tian et al., 2024).

**Step 2: Preference Optimization Using DPO.** After constructing $\mathcal{D}_{\text{pref}}$, we train our T2A model on this dataset with DPO using the approach outlined in Section 3. The resulting aligned model is referred to as $\mathcal{T}^{\theta}_{\text{aln}}$. Details of the hyper-parameters used for training are provided in Section 5.

## 4.2 Generating Diverse Synthetic Augmentations

It is not well-studied in the literature on how to leverage synthetic audio generation for downstream tasks. The only existing work relied on manually crafted prompt templates (e.g., "Sound of a {*label*}") (Ronchini et al., 2024). It has a significant limitation: there is no precise control over

---

[2] When prompted with "sound of a *bus*" for the category "*bus*" in the TUT-Urban dataset, the generated audio may not reflect the typical bus sounds in European cities (where TUT was recorded), as bus sounds can vary by region, with some featuring loud engines and dense crowds while others have quieter engines and sparse crowds.

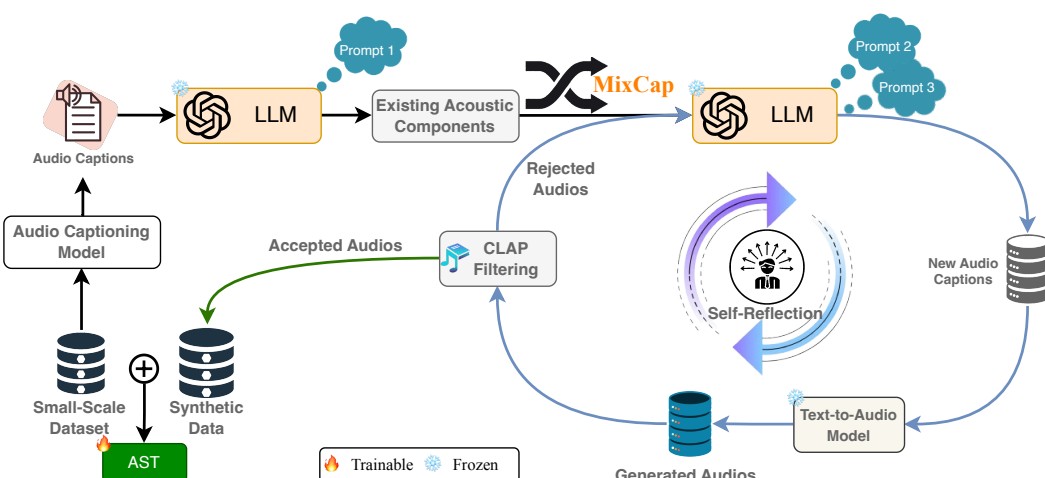

Figure 3: Overview of our proposed *Language-Guided Audio Imagination* for generating diverse synthetic augmentations. Starting with the small-scale dataset, we first generate audio captions and use an LLM to extract acoustic components (Prompt 1). Using these components and audio labels, we prompt the LLM to generate new and diverse captions (Prompt 2), which are then used to prompt the aligned T2A model for audio generation. The generated audios are filtered for label consistency using CLAP, with accepted audios added to the final synthetic dataset. Rejected audios undergo caption revision (Prompt 3) through a self-reflection process, and the revised captions are used to regenerate audios, iterating this process $i$ times. Example captions are in Table 6.

the specific components in the generated audio for a given caption. This can result in repetitive or completely inconsistent patterns, particularly with weaker T2A models [3]. These could bias the model to learn spurious correlations, a known issue in synthetic data augmentation (Ghosh et al., 2024c).

While the alignment stage helps the T2A model generate audio with acoustic characteristics similar to the small-scale dataset (e.g., spectral, harmonic, etc.), it does not fully account for the compositional diversity of the generated audios (e.g., sound events, their temporal relationships, background elements). To tackle this, we propose the concept of *language-guided audio imagination*, where we propose to imagine novel audios guided by language. Specifically, we leverage the reasoning abilities of LLMs to generate diverse and meaningful captions for a category label in a controlled yet scalable manner. These captions are then used to prompt our aligned T2A model for generating novel audios.

### 4.2.1 GENERATING DIVERSE PROMPTS WITH MIXCAP

We propose **MixCap**, a prompt generation method that creates diverse and effective captions in three steps: First, we employ GAMA (Ghosh et al., 2024a) to caption all audio files in $\mathcal{D}_{small}$. Next, we prompt an LLM to extract phrases describing the acoustic components of the audio. These components correspond to the acoustic elements such as backgrounds and foreground events, and their attributes and relations, etc. (see prompt in Appendix A.2). Finally, for each training instance in $\mathcal{D}_{small}$, we prompt the LLM with the ground-truth label and the extracted components from all instances to generate $N$ diverse audio captions that blend existing and new components.

### 4.2.2 FILTERING & SELF-REFLECTION

**Filtering.** After generating captions and their corresponding audio, we filter the audio for label consistency. While LLMs can generate diverse captions, the audio produced must remain aligned with the ground-truth label. To ensure this, we use CLAP to evaluate the generated audio, accepting those that meet a similarity threshold of $p\%$ and rejecting the rest. We denote the accepted audios as $\mathcal{D}_{syn}^{acc}$ and the rejected ones as $\mathcal{D}_{syn}^{rej}$. Our CLAP model is pre-trained on $\mathcal{D}_{a\text{-}c}$, and we fine-tune the last layer with $\mathcal{D}_{small}$ to adapt to the target dataset. Example captions are in Table 6.

**Self-Reflection.** For the rejected audios in $\mathcal{D}_{syn}^{rej}$, we prompt the LLM to reflect on its generated captions and revise them to better align with the target label. Precisely, we feed the LLM with the

---

[3]For example, when prompted with "Sound of a *park*", we observed that 9 out of 10 times, the model generated the sound of children playing as part of the generated audio. On the other hand, when prompted with "Sound of a *airport*", the model generates audios with background announcements, which could vary by region.

original caption of each rejected audio along with extracted components from all accepted captions in $\mathcal{D}_{\text{syn}}^{\text{acc}}$ and task it to rewrite the rejected captions. The revised captions are then used to generate new audio, which is again filtered using CLAP. Audios that meet the threshold are accepted, while ones that don't go through the process. This repeats for $i$ iterations or until there are no rejected samples.

**Fine-tuning for Audio Classification.** After the self-reflection stage, the final set of accepted synthetic audios is denoted as $\mathcal{D}_{\text{syn}}$, containing $\approx N \times n$ audio-label pairs, where $N$ represents the augmentation factor (e.g., with 100 gold samples, we generate $100 \times N$ synthetic samples). This is then combined with $\mathcal{D}_{\text{small}}$ to form the final training dataset $\mathcal{D}_{\text{train}}$ for audio classification.

## 5 EXPERIMENTAL SETUP

**Models and Hyper-Parameters.** For our T2A model, we choose the Stable Audio architecture (Evans et al., 2024b). We train the model from *scratch* on Sound-VECaps (Yuan et al., 2024) (with $\approx$1.5 million weakly captioned audio-caption pairs) to avoid any data leakage. For training, we employ a batch size of 64, an AdamW optimizer, a learning rate of 5e-4, and a weight decay of 1e-3 for 40 epochs. For DPO-based alignment tuning, we generate $j = 2$ losers and fine-tune with a batch size of 32 and a learning rate of 5e-4 for 12 epochs. For our audio classification model, we employ the Audio Spectrogram Transformer (AST) (Gong et al., 2021) (pre-trained on the AudioSet dataset) and fine-tune it with a batch size of 24 and a learning rate of 1e-4 for 50 epochs. For CLAP filtering, we employ $p = 0.85$. For prompting our diffusion model we use Text CFG=7.0. In each experiment, we adjust the number of generated augmentations $N$ (ranging from 1 to 5) based on performance on the validation set. All results are averaged across 3 runs.

**Datasets.** We create small-scale datasets by downsampling commonly used audio classification datasets to $n$ samples. Our selected datasets include a mix of music, everyday sounds, and acoustic scenes. For multi-class classification, we use NSynth Instruments, TUT Urban, ESC50 (Piczak), USD8K (Salamon et al., 2014), GTZAN (Tzanetakis et al., 2001), Medley-solos-DB (Lostanlen & Cella, 2017), MUSDB18 (Rafii et al., 2017), DCASE Task 4 (Mesaros et al., 2017), and Vocal Sounds (VS) (Mesaros et al., 2017), evaluating them for accuracy. For multi-label classification, we use the FSD50K (Fonseca et al., 2022) dataset and evaluate it using the $F_1^{macro}$ metric. We exclude AudioSet from evaluation as Sound-VECaps is derived from it. To ensure a downsampled dataset that has a label distribution similar to that of the of the original dataset, we employ stratified sampling based on categories. Our experiments are conducted with $n = \{50, 100, 200, 500\}$ samples, and we downsample the validation sets for training while evaluating all models on the original test splits.

**Baselines.** Our baselines include: (i) Gold-only (No Aug.): We employ only the small-scale dataset for training and do not perform any augmentations. (ii) Traditional augmentation baselines: SpecAugment, Noise Augmentation (we either add random Gaussian noise or background noise from AudioSet and present averaged results), Pitch and Time Shift and Audiomentations (Jordal, 2021) – a combination of the AddGaussianNoise, TimeStretch, PitchShift, Shift, SpecFrequencyMask, TimeMask and TimeStretch – combination with the highest average score on 4 datasets and splits and was selected after grid search over all possible combinations). (iii) Generative baselines: Vanilla Synthetic Augmentation (Vanilla Syn. Aug.) – we prompt $\mathcal{T}_\theta$ with template captions), Vanilla Syn. Aug. + LLM Caps – we prompt $\mathcal{T}_\theta$ with random captions generated with LLMs. (iv) Finally, inspired by Burg et al. (2023), we also employ a retrieval baseline where instead of generating augmentations from our T2A model trained on $\mathcal{D}_{\text{a-c}}$, we just retrieve the top-$n$ instances (w.r.t. CLAP similarity) from the AudioSet for each instance in $\mathcal{D}_{\text{small}}$ as our augmentations.

**Ablations.** We ablate Synthio with: (i)    w/o Self-Reflection: We remove the repetitive self-reflection module and iterate and filter only once; (ii)    w/o DPO: We skip the tuning step and prompt the un-alined $\mathcal{T}^\theta$ for augmentations; (iii)    w/ ERM: We replace DPO tuning with standard Empirical Risk Minimization(ERM)-based fine-tuning with diffusion loss; (iv)    w/ Template Captions: We remove MixCap and self-reflection modules and prompt $\mathcal{T}_{\text{aln}}^\theta$ with template captions; (v)    w/o MixCap: Similar to our Random Captions baseline, but we retain all other modules of Synthio.

## 6 RESULTS AND DISCUSSION

**Main Results.** Table 1 showcases the performance comparison between Synthio and the baseline methods. Synthio consistently outperforms all baselines by 0.1%-39%, achieving notable improve-

Table 1: Result comparison of Synthio with baselines on 10 datasets and 4 small-scale settings. $n$ refers to the number of samples in the small-scale dataset augmented with synthetic data. Synthio outperforms our baselines by 0.1% - 39%. We also highlight the relative improvements by Synthio compared to the Gold-only.

| $n$ | Method | ESC-50 | USD8K | GTZAN | Medley | TUT | NSynth | VS | MSDB | DCASE | FSD50K |
|---|---|---|---|---|---|---|---|---|---|---|---|
| | Gold-only (No Aug.) | 22.25 | 55.09 | 47.05 | 47.23 | 37.60 | 33.32 | 77.49 | 56.85 | 12.09 | 7.16 |
| | Random Noise | 18.50 | 57.42 | 45.20 | 46.55 | 35.86 | 32.42 | 76.41 | 52.55 | 13.21 | 8.06 |
| | Pitch Shifting | 20.55 | 59.32 | 46.80 | 48.17 | 37.22 | 34.34 | 78.17 | 54.50 | 12.93 | 10.04 |
| | SpecAugment | 19.50 | 58.36 | 46.00 | 47.18 | 36.73 | 27.32 | 77.27 | 53.25 | 12.81 | 7.93 |
| | Audiomentations | 20.35 | 60.13 | 47.25 | 48.30 | 38.24 | 28.15 | 79.12 | 54.51 | 13.28 | 10.17 |
| 50 | Retrieval | 19.20 | 37.14 | 42.55 | 43.65 | 35.80 | 31.27 | 71.42 | 51.35 | 10.53 | 7.28 |
| | Vanilla Syn. Aug. | 40.75 | 63.54 | 55.35 | 47.23 | 41.50 | 33.17 | 78.37 | 54.10 | 15.89 | 10.63 |
| | + LLM Caps. | 36.80 | 65.84 | 63.74 | 55.36 | 40.90 | 38.17 | 78.77 | 57.05 | 13.07 | 10.70 |
| | Synthio (ours) | $\mathbf{49.50}_{+122\%}$ | $\mathbf{76.12}_{+38\%}$ | $\mathbf{68.20}_{+44\%}$ | $\mathbf{60.58}_{+28\%}$ | $\mathbf{43.84}_{+17\%}$ | $\mathbf{40.83}_{+22\%}$ | $\mathbf{80.67}_{+4\%}$ | $\mathbf{60.15}_{+5\%}$ | $\mathbf{17.23}_{+42\%}$ | $\mathbf{13.91}_{+94\%}$ |
| | w/ Template Captions | 41.25 | 66.11 | 64.40 | 54.52 | 41.37 | 37.52 | 78.57 | 59.60 | 14.15 | 13.06 |
| | w/ ERM | 41.30 | 69.80 | 61.70 | 56.60 | 42.00 | 38.62 | 79.75 | 57.75 | 13.28 | 13.79 |
| | w/o Self-Reflection | 45.25 | 72.57 | 64.55 | 58.00 | 42.81 | 39.50 | 78.56 | 57.25 | 15.63 | 13.74 |
| | w/o MixCap | 42.70 | 64.72 | 54.65 | 52.18 | 41.93 | 36.13 | 78.70 | 58.80 | 14.82 | 12.52 |
| | w/o DPO | 36.55 | 68.12 | 56.10 | 52.55 | 41.39 | 40.31 | 79.03 | 57.55 | 14.53 | 10.13 |
| | Gold-only (No Aug.) | 56.75 | 72.89 | 64.15 | 57.81 | 47.14 | 39.11 | 84.32 | 65.60 | 12.50 | 10.53 |
| | Random Noise | 58.50 | 71.54 | 65.50 | 56.98 | 46.21 | 38.20 | 83.33 | 66.15 | 13.35 | 13.71 |
| | Pitch Shifting | 59.55 | 73.52 | 66.75 | 58.46 | 47.50 | 39.53 | 85.07 | 68.25 | 12.19 | 13.11 |
| | SpecAugment | 47.50 | 72.43 | 69.75 | 58.06 | 50.07 | 41.96 | 85.14 | 66.40 | 14.17 | 14.80 |
| | Audiomentations | 48.50 | 73.82 | 71.05 | 59.32 | 51.14 | 42.15 | 85.24 | 68.40 | 16.93 | 13.55 |
| 100 | Retrieval | 52.45 | 68.24 | 61.55 | 54.83 | 45.39 | 37.84 | 83.27 | 58.55 | 10.93 | 10.05 |
| | Vanilla Syn. Aug. | 77.25 | 77.31 | 68.25 | 63.58 | 49.96 | 42.31 | 84.78 | 63.55 | 15.73 | 12.63 |
| | + LLM Caps. | 67.05 | 79.73 | 67.90 | 65.79 | 48.63 | 41.83 | 84.83 | 65.95 | 16.32 | 13.25 |
| | Synthio (ours) | $\mathbf{83.35}_{+47\%}$ | $\mathbf{85.00}_{+17\%}$ | $\mathbf{71.20}_{+11\%}$ | $\mathbf{71.23}_{+23\%}$ | $\mathbf{52.42}_{+11\%}$ | $\mathbf{44.92}_{+15\%}$ | $\mathbf{86.70}_{+3\%}$ | $\mathbf{68.80}_{+5\%}$ | $\mathbf{19.38}_{+55\%}$ | $\mathbf{16.35}_{+55\%}$ |
| | w/ Template Captions | 78.00 | 80.32 | 68.15 | 64.20 | 49.95 | 42.76 | 85.11 | 66.05 | 16.32 | 13.62 |
| | w/ ERM | 73.20 | 81.81 | 67.25 | 66.57 | 51.11 | 43.74 | 84.73 | 68.00 | 17.21 | 14.52 |
| | w/o Self-Reflection | 77.65 | 82.38 | 69.55 | 68.52 | 51.75 | 44.38 | 82.53 | 66.20 | 15.89 | 12.14 |
| | w/o MixCap | 73.50 | 78.30 | 68.50 | 66.52 | 50.63 | 42.27 | 83.52 | 66.35 | 16.77 | 13.62 |
| | w/o DPO | 66.75 | 75.46 | 66.15 | 60.81 | 48.78 | 40.31 | 84.67 | 67.85 | 14.83 | 12.53 |
| | Gold-only (No Aug.) | 84.75 | 74.80 | 77.00 | 67.41 | 55.32 | 48.77 | 87.38 | 68.80 | 23.15 | 13.59 |
| | Random Noise | 83.55 | 75.15 | 75.50 | 66.71 | 54.42 | 47.83 | 86.45 | 65.45 | 24.82 | 15.32 |
| | Pitch Shifting | 84.90 | 74.48 | 78.55 | 67.74 | 55.44 | 48.12 | 87.47 | 69.80 | 23.11 | 17.51 |
| | SpecAugment | 85.10 | 76.46 | 76.25 | 65.70 | 55.72 | 54.80 | 87.42 | 69.25 | 27.36 | 17.93 |
| | Audiomentations | 85.25 | 75.80 | 77.30 | 67.00 | 55.21 | 53.15 | 86.08 | 70.50 | 26.29 | 18.36 |
| 200 | Retrieval | 82.55 | 71.20 | 73.65 | 65.80 | 53.25 | 47.63 | 86.28 | 63.55 | 19.51 | 15.36 |
| | Vanilla Syn. Aug. | 85.40 | 77.96 | 77.10 | 78.97 | 55.51 | 55.20 | 86.49 | 72.95 | 28.55 | 19.04 |
| | + LLM Caps. | 85.80 | 78.37 | 79.55 | 74.14 | 54.73 | 56.21 | 87.02 | 73.16 | 28.40 | 18.14 |
| | Synthio (ours) | $\mathbf{86.10}_{+2\%}$ | $\mathbf{82.81}_{+11\%}$ | $\mathbf{82.05}_{+7\%}$ | $\mathbf{79.40}_{+18\%}$ | $\mathbf{56.83}_{+3\%}$ | $\mathbf{57.10}_{+17\%}$ | $\mathbf{87.52}_{+0.2\%}$ | $\mathbf{80.40}_{+17\%}$ | $\mathbf{32.81}_{+42\%}$ | $\mathbf{20.85}_{+53\%}$ |
| | w/ Template Captions | 85.95 | 80.84 | 79.25 | 77.56 | 55.99 | 56.33 | 87.25 | 74.55 | 29.12 | 19.04 |
| | w/ ERM | 85.35 | 79.82 | 80.20 | 74.43 | 56.15 | 56.15 | 86.92 | 74.40 | 29.81 | 18.22 |
| | w/o Self-Reflection | 84.85 | 81.97 | 78.25 | 75.53 | 56.39 | 56.76 | 86.22 | 75.55 | 31.13 | 17.28 |
| | w/o MixCap | 84.95 | 81.27 | 79.55 | 73.50 | 55.27 | 55.54 | 85.78 | 78.55 | 28.35 | 19.42 |
| | w/o DPO | 84.80 | 76.23 | 75.30 | 73.13 | 55.99 | 52.73 | 86.52 | 73.15 | 26.79 | 17.17 |
| | Gold-only (No Aug.) | 90.75 | 87.88 | 79.25 | 75.65 | 65.72 | 63.47 | 89.33 | 72.05 | 34.30 | 20.19 |
| | Random Noise | 89.55 | 88.25 | 78.90 | 76.01 | 65.10 | 64.15 | 90.15 | 73.25 | 37.21 | 19.49 |
| | Pitch Shifting | 88.50 | 88.83 | 79.75 | 75.61 | 64.93 | 64.59 | 89.87 | 72.15 | 36.54 | 21.24 |
| | SpecAugment | 89.50 | 89.01 | 80.25 | 76.68 | 66.74 | 64.43 | 90.38 | 72.95 | 38.33 | 21.46 |
| | Audiomentations | 89.95 | 88.75 | 81.25 | 77.66 | 66.92 | 65.21 | 91.34 | 73.65 | 38.75 | 23.11 |
| 500 | Retrieval | 85.50 | 84.86 | 77.25 | 73.62 | 62.73 | 61.44 | 87.33 | 70.20 | 30.17 | 14.17 |
| | Vanilla Syn. Aug. | 91.50 | 88.18 | 79.35 | 77.97 | 65.93 | 64.52 | 90.31 | 73.25 | 37.26 | 23.52 |
| | + LLM Caps. | 89.90 | 86.91 | 79.55 | 77.91 | 65.95 | 64.39 | 90.09 | 73.05 | 38.74 | 22.67 |
| | Synthio (ours) | $\mathbf{92.10}_{+2\%}$ | $\mathbf{89.18}_{+2\%}$ | $\mathbf{82.25}_{+4\%}$ | $\mathbf{78.62}_{+4\%}$ | $\mathbf{67.81}_{+3\%}$ | $\mathbf{65.40}_{+3\%}$ | $\mathbf{91.42}_{+2\%}$ | $\mathbf{74.70}_{+3\%}$ | $\mathbf{39.24}_{+6\%}$ | $\mathbf{23.89}_{+18\%}$ |
| | w/ Template Captions | 91.70 | 88.93 | 80.40 | 76.64 | 66.47 | 64.71 | 90.97 | 73.35 | 38.28 | 22.35 |
| | w/ ERM | 91.20 | 88.25 | 79.15 | 77.38 | 65.80 | 64.27 | 88.74 | 74.20 | 38.03 | 22.39 |
| | w/o Self-Reflection | 91.85 | 88.72 | 80.15 | 78.57 | 66.21 | 63.89 | 90.17 | 72.15 | 37.97 | 22.41 |
| | w/o MixCap | 91.70 | 87.93 | 80.95 | 76.61 | 65.91 | 64.23 | 90.23 | 73.40 | 39.11 | 21.65 |
| | w/o DPO | 90.15 | 88.21 | 79.45 | 76.03 | 66.01 | 63.61 | 89.83 | 72.65 | 37.04 | 20.19 |

ments in overall classification accuracy compared to Gold-only. The highest gains are observed on USD8K, while the least is on Vocal Sound, likely due to the T2A dataset's heavy representation of music compared to the more sparse vocal sounds. Performance gains tend to decrease as the number of gold samples $n$ in $\mathcal{D}_{\text{small}}$ grows, aligning with observed trends in prior studies. Detailed results on the full non-down-sampled datasets can be found in Appendix A.4.1. Although Vanilla Synthetic Augmentations emerge as the strongest baseline, they lag behind Synthio by an average of 3.5%.

**Ablations.** The most significant performance drop in Synthio is observed w/o DPO, resulting in an average decline of 4.5%, highlighting the crucial role of consistency in generating effective augmentations. Second to w/o DPO, the highest drop is seen in w/ Template Captions, with an average decline of 2.7%, thus highlighting the importance of MixCap.

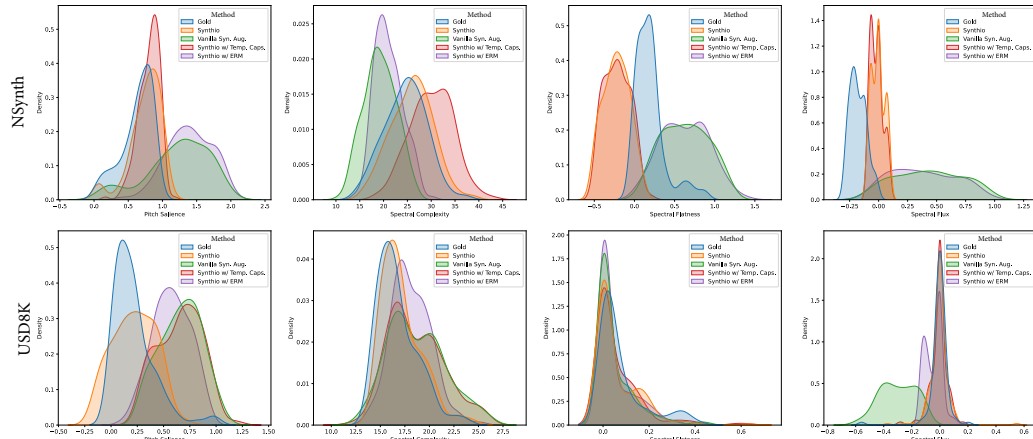

Figure 4: Comparison of spectral and pitch features between generated audios in $\mathcal{D}_{\text{syn}}$ and real audios in $\mathcal{D}_{\text{small}}$ (for $n = 100$). Synthio-generated audios closely replicate the features of real data, demonstrating its ability to produce augmentations that maintain consistency with the original dataset (also see FAD scores in Sec. A.4.3).

## 6.1 HOW CONSISTENT AND DIVERSE ARE AUGMENTATIONS GENERATED BY SYNTHIO?

Fig. 4 (analysis inspired from Cherep & Singh) compares the distributions of pitch and various spectral features between generated audios in $\mathcal{D}_{\text{syn}}$ and real audios in $\mathcal{D}_{\text{small}}$ across different methods on the USD8K and NSynth datasets. The features analyzed include Pitch Salience (clarity of the main pitch) (Ricard, 2004), Spectral Flatness (tonal vs. noise-like quality) (Peeters, 2004), Flux (rate of spectral change) (Tzanetakis & Cook, 1999), and Complexity (level of sound detail) (Laurier et al., 2010). Notably, Synthio-generated audios closely replicate the spectral features of the original audios, showing the best alignment among all methods and demonstrating Synthio's ability

Table 2: CLAP similarity score between real audios and generated data. Lower scores show higher compositional diversity among generated augs.

| # | Method | USD8K(↓) | NSynth(↓) |
|---|---|---|---|
| 100 | Vanilla Syn. Aug. | 45.17 | 31.76 |
| | Synthio *(ours)* | **35.09** | **22.97** |
| | w/ Template Captions | 46.82 | 33.00 |
| | w/ ERM | 50.01 | 42.33 |
| 200 | Vanilla Syn. Aug. | 47.22 | 33.81 |
| | Synthio *(ours)* | **34.58** | **23.03** |
| | w/ Template Captions | 46.84 | 37.16 |
| | w/ ERM | 52.54 | 43.98 |

to generate consistent augmentations. Table 2 presents CLAP similarity scores between ground-truth audios and their $N$ generated augmentations, averaged across all dataset instances. Audios generated with Synthio achieve the highest compositional diversity for generated audios among all baselines. Table 8 shows that audios generated using Synthio have the highest similarity with the ground-truth category label.

## 6.2 HOW GOOD ARE SYNTHETIC AUDIOS GENERATED BY SYNTHIO?

Consistent with prior findings in vision (He et al., 2023), we observe that synthetic data alone performs sub-optimally compared to human-annotated data. However, our results show that enhancing the consistency and diversity of synthetic data aided by a small-scale version of the target dataset significantly improves model performance. Table 3 compares models trained exclusively on synthetic data with our baselines (i.e., only $\mathcal{D}_{\text{syn}}$ is used for training AST). Synthio outperforms all baselines by 0.1%-26.25%, with DPO-based alignment driving the improvements.

Table 3: Performance comparison of Synthio with baselines on *synthetic-only* audio classification.

| $n$ | Method | GTZAN | VS | TUT | MSDB |
|---|---|---|---|---|---|
| | Gold-only (No Aug.) | 64.15 | 84.32 | 47.14 | 65.60 |
| 100 | Vanilla Syn. Aug. | 29.05 | 34.13 | 21.69 | 35.60 |
| | Synthio *(ours)* | **33.10** | **39.20** | **24.51** | **56.45** |
| | w/ Template Captions | 24.50 | 30.99 | 21.73 | 40.40 |
| | w/ ERM | 25.65 | 32.76 | 24.40 | 42.85 |
| | w/o DPO | 17.60 | 21.57 | 20.39 | 30.20 |
| | Gold-only (No Aug.) | 77.00 | 87.38 | 55.32 | 68.80 |
| 200 | Vanilla Syn. Aug. | 32.35 | 41.96 | 24.23 | 39.25 |
| | Synthio *(ours)* | **35.15** | **48.14** | **27.00** | **61.45** |
| | w/ Template Captions | 29.90 | 35.53 | 23.61 | 41.20 |
| | w/ ERM | 28.10 | 36.29 | 25.71 | 46.70 |
| | w/o DPO | 19.85 | 26.85 | 21.40 | 36.75 |

## 6.3 CAN SYNTHIO BE EXTENDED TO THE MORE COMPLEX AUDIO CAPTIONING TASK?

Audio captioning, unlike classification, involves describing the content of an audio sample using natural language, making it a more complex task. To demonstrate Synthio's effectiveness for audio captioning, we evaluated it on down-sampled versions of Audio-Caps. For this task, we adapted Synthio by removing the audio captioning and CLAP filtering stages, and we extracted acoustic features directly from the existing audio captions.

Additionally, we retrain our T2A model on a modified version of Sound-VECaps, excluding any audio from AudioCaps. Training and evaluation were conducted using the EnCLAP framework (Kim et al., 2024), and the dataset was expanded with 4× synthetic samples. As shown in Table 4, Synthio significantly outperforms baseline settings, with improvements largely due to better alignment w/ DPO. However, manual

Table 4: Performance comparison of Synthio with baselines on audio captioning.

| $n$ | Method | METEOR (↑) | CIDEr (↑) | SPICE (↑) | SPIDEr (↑) |
|---|---|---|---|---|---|
| 500 | Gold-only (No Aug.) | 0.125 | 0.148 | 0.0754 | 0.112 |
| | Vanilla Syn. Aug. | 0.128 | 0.157 | 0.0741 | 0.136 |
| | VECaps Retrieval | 0.108 | 0.094 | 0.0550 | 0.082 |
| | Synthio (ours) | **0.169** | **0.227** | **0.104** | **0.194** |
| 1000 | Gold-only (No Aug.) | 0.127 | 0.157 | 0.067 | 0.112 |
| | Vanilla Syn. Aug. | 0.135 | 0.166 | 0.092 | 0.140 |
| | VECaps Retrieval | 0.088 | 0.097 | 0.068 | 0.100 |
| | Synthio (ours) | **0.185** | **0.256** | **0.119** | **0.202** |

inspection revealed that generated audios occasionally do not match their captions compositionally, reflecting limitations of the current T2A model. While this issue does not affect classification, it poses challenges for captioning. We will explore more advanced methods as part of future work.

### 6.4 HOW WELL DOES SYNTHIO SCALE?

Table 5 compares the performance of Synthio, SpecAugment, and Vanilla Synthetic Augmentations across different scaling factors $N = \{1, 2, 3, 4, 5\}$, where $N$ represents the number of synthetic samples generated per original sample in the small-scale dataset (in this case we fix $n = 100$). As observed, SpecAugment, a traditional augmentation method, cannot scale with increasing $N$, and the performance of Vanilla plateaus at higher $N$. A similar saturation occurs with Synthio when MixCap is not used. Even without DPO, Synthio maintains better scalability, though with reduced overall performance. These results

Table 5: Performance comparison of Synthio with other baselines on different values of $N$.

| Dataset | Method | Scaling Factor $N$ | | | | |
|---|---|---|---|---|---|---|
| | | 1x | 2x | 3x | 4x | 5x |
| ESC50 | SpecAugment | 47.50 | 47.50 | 47.50 | 47.50 | 47.50 |
| | Vanilla Syn. Aug. | 67.90 | 77.25 | 76.75 | 75.60 | 71.25 |
| | Synthio (ours) | 77.45 | 81.75 | 82.55 | 83.15 | **83.35** |
| | w/o MixCap | 64.30 | 68.45 | 71.55 | 72.85 | 73.50 |
| | w/o DPO | 61.55 | 64.25 | 65.95 | 66.60 | 66.75 |
| NSynth | SpecAugment | 41.96 | 41.96 | 41.96 | 41.96 | 41.96 |
| | Vanilla Syn. Aug. | 33.13 | 35.28 | 42.31 | 41.54 | 38.27 |
| | Synthio (ours) | 35.28 | 36.37 | 43.56 | **44.92** | 44.81 |
| | w/o MixCap | 40.41 | 41.08 | 41.95 | 42.27 | 42.15 |
| | w/o DPO | 39.23 | 39.42 | 40.17 | 40.31 | 39.82 |

highlight that MixCap's ability to generate diverse captions is crucial for Synthio's scalability.

### 6.5 DOES SYNTHIO HELP LONG-TAILED CATEGORIES?

Figure 5 shows the classification accuracy on four underrepresented categories in the NSynth dataset, comparing performance before and after applying Synthio augmentations. We selected categories with the lowest frequency in the downsampled dataset, such as *flute* and *guitar*, which appear only once in the down-sampled sets. Synthio significantly boosts accuracy, with improvements up to 48%. Notably, category labels like *flute* and *guitar*, which originally had 0% accuracy, show substantial gains with Synthio augmentation. This

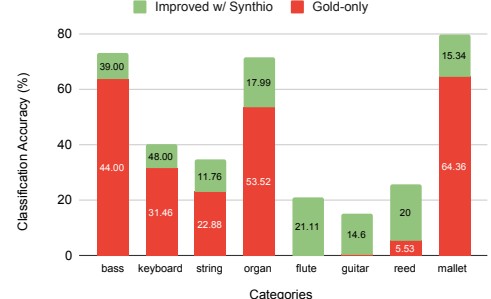

Figure 5: Category-wise improvement in performance with Synthio augmentations for long-tailed categories.

demonstrates Synthio's effectiveness in boosting performance on long-tail labels, a common challenge in real-world datasets (Zhang et al., 2023).

## 7 CONCLUSION, LIMITATIONS, AND FUTURE WORK

We introduced Synthio, a novel approach for augmenting small-scale audio classification datasets with synthetic data. Synthio incorporates several innovative components to generate augmentations that are both consistent with and diverse from the small-scale dataset. Our extensive experiments demonstrate that even when using a T2A model trained on a weakly-captioned AudioSet, Synthio significantly outperforms multiple baselines.

However, Synthio has some limitations: (i) Its performance is influenced by the capabilities of the T2A model and the quality of its training data. As T2A models continue to improve, we expect Synthio's performance to benefit accordingly. (ii) The process of generating audio captions using LLMs may introduce biases inherent in the LLMs into the training process. (iii) Synthio is computationally more intensive than traditional augmentation methods due to the need for prompting LLMs and T2A models. We anticipate that ongoing advancements in model efficiency will help mitigate these computational challenges.

# 8 REPRODUCIBILITY STATEMENT

Our project page has all the codes and checkpoints to reproduce the results in the paper. All experimental details, including training parameters and hyper-parameters, are provided in Section 5.

# 9 ACKNOWLEDGMENT

This project is supported in part by NSF#1910940.

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

## A APPENDIX

**Table of Contents:**

### A.1 DIFFUSION MODELS

Diffusion models consist of two main processes: a forward process and a reverse process. Given a data point $x_0$ with probability distribution $p(x_0)$, the forward diffusion process gradually adds Gaussian noise to $x_0$ according to a pre-set variance schedule $\beta_1, \cdots, \beta_T$ and degrades the structure of the data. At the time step $t$, the latent variable $x_t$ is only determined by the $x_{t-1}$ due to its discrete-time Markov process nature, and can be expressed as:

$$p(x_t \mid x_{t-1}) = \mathcal{N}(x_t; \sqrt{1 - \beta_t}x_{t-1}, \beta_t I), \tag{12}$$

As $t$ increases over several diffusion steps, $p(x_T)$ approaches a unit spherical Gaussian distribution. The marginal distribution of $x_t$ at any given step can be expressed analytically as:

$$p(x_t \mid x_0) = \mathcal{N}(x_t; \sqrt{\alpha_t}x_0, (1 - \alpha_t)I), \tag{13}$$

where $\alpha_t = \prod_{s=1}^{t}(1 - \beta_s)$. The reverse process aims to reconstruct the original data from the noise-corrupted version by learning a series of conditional distributions. The transition from $x_t$ to $x_{t-1}$ is modeled as:

$$p_\theta(x_{t-1} \mid x_t) = \mathcal{N}(x_{t-1}; \mu_\theta^{t-1}, \sigma_\theta^{t-1}), \tag{14}$$

$$\mu_\theta^{t-1} = \frac{1}{\sqrt{\alpha_t}}\left(x_t - \frac{\beta_t}{\sqrt{1 - \bar{\alpha}t}}\epsilon_\theta\left(x_t, t\right)\right), \tag{15}$$

$$\sigma_\theta^{t-1^2} = \frac{1 - \bar{\alpha}_{t-1}}{1 - \bar{\alpha}_t} \cdot \beta_t, \tag{16}$$

where $\alpha_t = 1 - \beta_t$, $\bar{\alpha}_t = \prod_{i=1}^{t} \alpha_i$, $\theta$ represents the learnable parameters, $\mu_\theta^{t-1}$ is the mean estimate, $\sigma_\theta^{t-1^2}$ is the standard deviation estimate, and $\epsilon_\theta(x_t, t)$ is the noise estimated by the neural network. The reverse process estimates the data distribution $p(x_0)$ by integrating over all possible paths:

$$p_\theta(x_0) = \int p_\theta(x_T) \prod_{t=1}^{T} p_\theta(x_{t-1} \mid x_t) \, dx_1 : T \tag{17}$$

where $p_\theta(x_T) = \mathcal{N}(x_T; 0, I)$. At inference time, the diffusion model iteratively executes the reverse process (Eq. 17) $T$ times starting from a randomly sampled Gaussian Noise ($\epsilon \sim \mathcal{N}(0, \mathbf{I})$).

At the time step $t$, the latent variable $x_t$ is only determined by the $x_{t-1}$ due to its discrete-time Markov process nature. At inference time, the diffusion model iteratively executes the reverse process $T$ times starting from a randomly sampled Gaussian Noise ($\epsilon \sim \mathcal{N}(0, \mathbf{I})$). For more details on diffusion models, we request our readers to refer to Appendix A.1.

### A.2 PROMPTS

Fig. 6, 7, 8 and 9 illustrate all the prompts used in our experiments. For all experiments, we prompt GPT-4-Turbo (GPT-4-turbo-2024-04-09) with top-p=0.5 and temperature=0.7.

### A.3 EXAMPLES

Table 6 presents examples of captions generated by the Synthio framework, along with their revised versions for captions that were initially rejected.

```
I will provide you with a caption of an audio that describes the events taking place in the
audio. Additionally, I will also provide you with a label for the audio. Extract the phrases
that correspond to the distinctive features of the audio. There are 3 types of features you need
to extract:
1) the unique foreground events in the caption,
2) the broader background scene or background events in the or audio and
3) any other features related to the audio. Return a JSON with key 3 keys, one as named as
'events', the other as named as 'scenes', and the other named as 'other features', where the
values of these keys correspond to a comma-separated pythonic list where each item in the list
is a string corresponding to the extracted phrases. Please ignore any phrase that (exactly or
semantically) corresponds to the label of the audio. If you think there is no information for
either of the keys, leave them empty.

Here is the caption:{}
Here is the label:{}
```

Figure 6: LLM prompt (Prompt 1) for extracting components from audio captions.

```
I will provide you with a caption for an audio. The label generally describes the audio in an
abstract fashion and mentions the broader scene or event that I need to teach an audio model
about from the audio, i.e., the audio and its label is part of the training set for training an
audio classification model. I will also provide you with the domain of the audio which will help
you identify the true sound conveyed in the label. I need you to rewrite the caption for me
according to this set of rules:
1. I will provide you with lists of various audio features corresponding to events, backgrounds
or other features. You should rewrite the given caption such that it has has features inspired
from the features provided to you, i.e., you should try to describe a scene for the label with
events, backgrounds and features similar but unique from the ones given.
2. After re-writing, the caption should still obey the audio event label.

Here is the label:{}.

Here is the domain of the audio:{}.
Here is the list of events:{}
Here is the list of backgrounds:{}
Here is the list of other features:{}

Just output the rewritten caption and nothing else. Output 'None' if you did not rewrite.
```

Figure 7: LLM prompt (Prompt 2) for generating new audio captions given elements from existing captions.

## A.4 EXTRA RESULTS

### A.4.1 RESULTS ON THE FULL TRAINING SPLITS

Table 7 presents the performance comparison of Synthio on the full original dataset splits (where the entire training set is used without any downsampling). While Synthio outperforms all baselines, traditional augmentation methods prove to be much more competitive in this scenario. This contrasts with the results in Table 1 where traditional augmentations showed minimal improvements in performance.

**Additional Discussion on Results.** As we see in Table 1 (and Table 7), performance gains with Synthio as the number of Gold samples increase (highest absolute gains with $n = 100$ and lowest with full dataset). This phenomenon is consistent across prior work in vision (Trabucco et al., 2024), text (Ghosh et al., 2023a; 2024c), and audio (Ronchini et al., 2024). Most synthetic data augmentation methods demonstrate substantial gains in low-resource regimes, but these gains naturally diminish as the quantity of high-quality labeled data increases (for example, Azizi et al. just show over ImageNet only a modest improvement of just over 1%, where the authors reported when augmenting this large-scale dataset).

We hypothesize that this trend is rooted in the inherent diversity and richness of gold data. Gold datasets typically capture nuanced variations and complex real-world distributions, including subtle contextual and environmental factors that synthetic data struggles to replicate. Synthetic data, while effective at filling gaps and addressing low-resource scenarios, often lacks the granularity necessary to represent long-tail or edge-case instances. As the size of the gold dataset increases, the model

```
I will provide you with a label for an audio. The label generally describes the audio in an
abstract fashion and mentions the broader scene or event that I need to teach an audio model
about from the audio, i.e., the audio and its label is part of the training set for training an
audio classification model. I will also provide you with the domain of the audio which will help
you identify the true sound conveyed in the label. I would like you to generate 5 new captions
that describe the event or source in the label in diverse fashions. I will use these captions to
generate new audios that can augment my training set. Generate the new captions with the
following requirements:

1. All the captions need to include new and diverse events and contexts beyond the actual event
conveyed by the label.
2. Only add new events and context by understanding the broader context of the occurrence of the
audio and the target label. Do not add random events or contexts.
3. The new caption should be not more than 20-25 words.
4. However, after all these constraints and adding new events or contexts, the caption still
needs to obey the event conveyed by the original label, i.e., the new caption may not lead to an
audio generation that defies the audio label.
6. Finally, use the original label as a phrase in your caption.

Here is the label:{}.

Here is the domain of the audio:{}. Output a JSON with the key as the original label and a value
as the list of comma separated new captions. Only output the JSON and nothing else
```

Figure 8: LLM prompt for generating random captions for Random Captions baselines in Table 1.

```
I will provide you with a label for an audio. The label generally describes the audio in an
abstract fashion and mentions the broader scene or event that I need to teach an audio model
about from the audio, i.e., the audio and its label is part of the training set for training an
audio classification model. I will also provide you with the domain of the audio which will help
you identify the true sound conveyed in the label. I would like you to generate 5 new captions
that describe the event or source in the label in diverse fashions. I will use these captions to
generate new audios that can augment my training set. Generate the new captions with the
following requirements:

1. Each caption should have a diverse added events (beyond the event of the original label) and
contexts.
2. Only add new events and context by understanding the broader context of the occurrence of the
audio and the target label. For adding events and contexts, please follow the next requirement.
3. I will also provide you with a list of features extracted from an existing set of audios. You
should try such that the new captions you generate for the label have a mix of events and scenes
similar to the events and background scenes that are given and new scenes, i.e., you should try
to describe a scene for the caption with the events and backgrounds provided to you in the given
lists but you should also add novel features (events, backgrounds or other features) beyond the
ones given.
4. The new caption should be not more than 20-25 words.
5. However, after all these constraints and adding new events or contexts, the caption still
needs to obey the event label, i.e., the new caption may not lead to an audio generation that
defies the audio label.
6. Finally, use the original label as a phrase in your caption.

Here is the label:{}.

Here is the domain of the audio:{}.
Here is the list of events:{}
Here is the list of backgrounds:{}
Here is the list of other features:{}

Output a JSON with the key as the original caption and a value as the list of comma separated
new captions. Only output the JSON and nothing else.
```

Figure 9: LLM prompt (Prompt 3) for rewriting captions of rejected audios.

increasingly benefits from the inherent diversity of these high-quality examples, reducing the need for synthetic data and its relative impact on performance.

Additionally, in Fig. 6 of their paper, Azizi et al. also how an increasing number of synthetic augmentations leads to plateauing and even diminishing performance. We hypothesize that this is due to over-fitting caused by lack of diversity in generated augmentations.

| Dataset | Label | Generated Caption | Revised Caption |
|---------|-------|-------------------|-----------------|
| USD8k | children_playing | Children playing in a bustling city park with distant traffic noise | NA |
| USD8k | children_playing | Children playing in a schoolyard during recess with teacher's whistle | Children playing in a neighborhood alley with sound of distant construction |
| USD8k | street_music | Street music playing near a busy intersection filled with honking cars and pedestrians. | NA |
| USD8k | street_music | Street music from a bustling market as people chatter and vendors shout | Street music echoing through an alleyway during a lively street festival. |
| TUT | airport | airport with people talking and walking around in an empty hallway | NA |
| TUT | airport | In the airport, people are talking with the sound of a crowd of people in the background, as announcements play. | airport ambiance with people talking and children running around |
| TUT | bus | Bus passing by on a road while people are chatting at a nearby cafe. | NA |
| TUT | bus | bus passing by on a road as it continues to blow into the microphone | bus idling on a road with birds chirping nearby |
| NSynth | keyboard | keyboard accompaniment to a live band performance at a bustling cafe. | NA |
| NSynth | keyboard | a man typing on a keyboard at office | keyboard rhythms echoing in an empty auditorium during a rehearsal break |
| NSynth | organ | A serene church service with an organ playing a melody and soft brass are playing. | NA |
| NSynth | organ | An organ plays as guitars are playing together in the background. | An organ plays during a lively music festival with various instruments. |
| Medley | Violin | violin being played during a classical symphony orchestra performance | NA |
| Medley | Violin | violin performing a lively jig at a bustling street fair | Violin solo during a quiet candlelight dinner in a fancy restaurant. |
| Medley | Flute | flute playing in a tranquil forest during the early morning | NA |
| Medley | Flute | Flute performance in a bustling city park during a sunny afternoon. | Flute music echoing in an ancient stone cathedral. |
| AudioCaps | - | A dog barks repeatedly in the background while a car engine starts | - |
| AudioCaps | - | In the distance, a faint thunder rumble is audible, accompanied by the gentle rustling of leaves in the wind. | Soft rain falls on a metal roof, creating a rhythmic tapping sound. |

Table 6: Examples of generated and revised captions from the Synthio methodology.

Table 7: Comparison of Synthio and other baselines on the full original dataset splits (using all samples from the original training set as $\mathcal{D}_{\text{small}}$).

| Method | USD8K | GTZAN | Medley | VS | MSDB |
|--------|-------|-------|--------|-----|------|
| Gold-only | 88.23 | 82.00 | 80.99 | 92.73 | 73.9 |
| Random Noise | 86.17 | 82.35 | 79.72 | 92.94 | 74.55 |
| Pitch Shift | 87.58 | 83.02 | 79.63 | 92.17 | 74.6 |
| Spec. Aug. | 87.92 | 82.50 | 79.14 | 92.42 | 74.5 |
| Audiomentations | 88.01 | 82.75 | 81.26 | 92.47 | 75.05 |
| Retrieval | 78.27 | 69.25 | 73.24 | 80.43 | 69.95 |
| Vanilla Syn. Aug. | 89.57 | 82.85 | 81.79 | 93.15 | 75.85 |
| Synthio *(ours)* | **89.57** | **82.85** | **81.79** | **93.01** | **74.24** |

### A.4.2 AUDIO GENERATION RESULTS FOR OUR TRAINED STABLE DIFFUSION

Table 9 presents a comparison of audio generation results across several evaluation metrics. We evaluate our trained Stable Diffusion model (used in our experiments, including a version further

Table 8: CLAP score between generated audios and the label.

| $n$ | Method | USD8K | NSynth |
|-----|--------|-------|--------|
| 100 | Real | 12.67 | 14.46 |
| | Vanilla Syn. Aug. | 14.34 | 17.54 |
| | Synthio | **31.26** | **27.32** |
| | w/ Template Captions | 29.31 | 26.62 |
| | w/ ERM | 24.15 | 21.54 |
| 200 | Real | 10.13 | 9.4 |
| | Vanilla Syn. Aug. | 12.55 | 12.91 |
| | Synthio | **21.87** | **16.16** |
| | w/ Template Captions | 20.31 | 15.82 |
| | w/ ERM | 17.14 | 13.04 |

Table 9: Comparison of our trained Stable Diffusion model on AudioCaps test set

| Model | FAD_PANN (↓) | FAD_VGG (↓) | IS_PANN (↑) | CLAP_LAION (↑) |
|-------|-------------|-------------|-------------|----------------|
| AudioLDM2-large | 32.50 | 1.89 | 8.55 | 0.45 |
| Tango-Full0FT-AC | 18.47 | 2.19 | 8.80 | 0.57 |
| Tango 2 | 17.19 | 2.54 | 11.04 | 0.52 |
| Make-an-Audio 2 | 11.75 | **1.80** | - | **0.60** |
| Stable Audio VECaps (ours) | 15.12 | 2.21 | 15.07 | 0.57 |
| Stable Audio VECaps + AudioCaps-FT *(ours)* | **14.93** | 2.19 | **15.42** | 0.56 |

fine-tuned on AudioCaps) against other available models and baselines from the literature. Notably, our model performs competitively with other fully open-source models across most metrics.

### A.4.3 FAD SCORES FOR GENERATED AUGMENTATIONS

To offer an alternative perspective on the distributional consistency between the generated augmentations and the ground-truth small-scale dataset, we compare the Fréchet Audio Distance (FAD) scores (Kilgour et al., 2018). For this experiment, we use Synthio with Template Captions. Table 10 presents a comparison of FAD scores between Synthio and other baselines. Synthio achieves the highest FAD score, indicating that it produces the most consistent audio augmentations.

Table 10: Comparison of FAD score of Vaniall Syn. Aug. and Stable Audio VECaps (ours).

| $n$ | Dataset | Model | FAD_VGG (↓) |
|-----|---------|-------|-------------|
| 100 | NSynth | Vanilla Syn. Aug. | 1.83 |
| | | Stable Audio VECaps (ours) | **1.42** |
| 200 | TUT | Vanilla Syn. Aug. | 1.71 |
| | | Stable Audio VECaps (ours) | **1.45** |

### A.4.4 EFFECT OF CLAP FILTERING

In this section, we provide additional experiments to show the effect of CLAP filtering on the Synthio pipeline. Table 11 compares the performance of Synthio with and without CLAP. As we can see,

Table 12 compares the performance of various values of $p$ on 5 datasets and 2 values of $n$ (500 and 100). As we see, higher or lower values of $p$ do not affect the final performance significantly.

Our T2A model uses the same CLAP text encoder for generating audio. Consequently, most generated audios are already highly aligned with the intended category label. However, the purpose of CLAP filtering is to safeguard against cases where the LLM hallucinates and generates a caption that deviates significantly from the intended label. In such cases, CLAP filtering ensures that audios generated from hallucinated captions are discarded, preventing them from negatively impacting the learning process.

Table 11: Ablation study evaluating the impact of CLAP filtering on Synthio's performance.

| n | Method | ESC-50 | USD8K | GTZAN | TUT | VS |
|---|--------|--------|-------|-------|-----|-----|
| 50 | Synthio | **49.50** | **76.12** | **68.20** | **43.84** | **80.67** |
| | Synthio w/o CLAP | 47.25 | 74.34 | 66.35 | 40.28 | 77.29 |
| 100 | Syhtio | **83.35** | **85.00** | **71.20** | **71.23** | **86.70** |
| | Synthio w/o CLAP | 82.55 | 84.64 | 69.30 | 70.41 | 84.93 |
| 200 | Syhtio | **86.10** | **82.81** | **82.05** | **56.83** | **87.52** |
| | Synthio w/o CLAP | 85.25 | 79.94 | 80.54 | 55.22 | 86.31 |
| 500 | Syhtio | **92.10** | **89.18** | **82.25** | **67.81** | **91.42** |
| | Synthio w/o CLAP | 90.25 | 88.42 | 79.70 | 65.42 | 89.67 |

Table 12: Comparison of Synthio's performance with different CLAP threshold levels.

| n | p | ESC-50 | USD8K | GTZAN | TUT | VS |
|---|---|--------|-------|-------|-----|-----|
| 50 | 0.85 | **49.50** | **76.12** | **68.20** | **43.84** | **80.67** |
| | 0.3 | 47.10 | 74.14 | 67.50 | 41.17 | 79.32 |
| | 0.5 | 48.25 | 75.39 | 67.75 | 41.93 | 79.48 |
| 100 | 0.85 | **83.35** | **85.00** | **71.20** | **71.23** | **86.70** |
| | 0.3 | 82.55 | 84.64 | 69.30 | 70.41 | 84.93 |
| | 0.5 | 82.70 | 84.73 | 70.25 | 70.86 | 85.22 |
| 200 | 0.85 | **86.10** | **82.81** | **82.05** | **56.83** | **87.52** |
| | 0.3 | 85.25 | 79.94 | 80.55 | 55.22 | 86.31 |
| | 0.5 | 85.70 | 80.30 | 81.30 | 56.19 | 87.11 |
| 500 | 0.85 | **92.10** | **89.18** | **82.25** | **67.81** | **91.42** |
| | 0.3 | 90.25 | 88.42 | 80.70 | 65.42 | 89.67 |
| | 0.5 | 91.65 | 89.07 | 81.05 | 66.35 | 90.02 |

### A.4.5 EFFECT OF TRAINING DATA AND MODEL ARCHITECTURE FOR THE TEX-TO-AUDIO MODEL

In this section, we train our T2A model using 1) a different model architecture (we replace Stable Diffusion with Tango Ghosal et al. (2023)) different training data (we replaced Sound-VECaps with AudioCaps). Table 13 compares thee results. As we can clearly see, while the model architecture of the T2A model does not affect the performance, replacing the training data with a small and less diverse dataset leads to significant drop in performance.

Table 13: Comparison of Synthio with Synthio's Stable Audio trained only wiht AudioCaps and Tango trained with Sound-VECaps

| n | Method | ESC-50 | USD8K | GTZAN | Medley | TUT |
|---|--------|--------|-------|-------|--------|-----|
| 50 | Synthio *(ours)* | **49.50** | **76.12** | **68.20** | **60.58** | **43.84** |
| | Synthio w/ AudioCaps | 29.20 | 60.15 | 50.15 | 49.19 | 38.62 |
| | Synthio w/ Tango | 48.55 | 75.05 | 66.19 | 59.12 | 42.59 |
| 100 | Synthio *(ours)* | **83.35** | **85.00** | **71.20** | **71.23** | **52.42** |
| | Synthio w/ AudioCaps | 58.20 | 74.27 | 66.55 | 67.93 | 48.23 |
| | Synthio w/ Tango | 81.50 | 84.13 | 70.95 | 69.97 | 51.47 |

### A.4.6 SYNTHIO AS A COMPLIMENTARY APPROACH TO TRADITIONAL AUGMENTATIONS

Table 14 compares results of Synthio augmentations when combined with traditional augmentations. As we can see, Synthio boosts performance of all methods and combining traditional augmentations with Synthio boosts Synthios overall performance. This shows that Synthio can act as a complimentary step for traditional augmentations.

**Additional Discussion.** Across all datasets, we noticed that CLAP filtering removed at most 10% of the generated samples. This confirms that the majority of the synthetic data is already well-aligned with the target categories, and filtering primarily handles rare cases of misalignment. Thus we emphasize on the point that while most generated audios align with the target label, the CLAP filtering stage acts as a safeguard against hallucinations by the LLM, which may occasionally generate

Table 14: Performance comparison of Synthio when paired with traditional augmentation techniques

| $n$ | Method | ESC-50 | USD8K | GTZAN | Medley |
|---|---|---|---|---|---|
| 50 | Synthio (ours) | 49.50 | 76.12 | 68.20 | 60.58 |
| | w/ Random Noise | 49.65 | 77.31 | 70.15 | 61.54 |
| | w/ Pitch Shift | 49.80 | **78.52** | 69.50 | 60.29 |
| | w/ Spec Aug | **50.95** | 77.93 | **70.35** | 61.17 |
| | w/ Audiomentations | 50.35 | 77.24 | 69.50 | **61.53** |
| 100 | Synthio (ours) | 83.35 | 85.00 | 71.20 | 71.23 |
| | w/ Random Noise | 83.85 | **86.59** | 71.60 | 72.35 |
| | w/ Pitch Shift | 83.60 | 86.32 | **72.95** | 72.50 |
| | w/ Spec Aug | **84.25** | 86.17 | 72.75 | **73.05** |
| | w/ Audiomentations | 84.10 | 85.95 | 72.85 | 72.87 |

captions that deviate significantly from the intended category. In such cases, filtering ensures that misaligned audios are discarded, preventing them from negatively impacting model training.

## A.5 DATASET DETAILS

**NSynth Instruments:** NSynth is a large-scale dataset consisting of musical notes played by a variety of instruments. It includes a rich set of acoustic features from instruments like guitars, flutes, and more, providing diverse sound textures for classification tasks.

**TUT Urban:** The TUT Urban dataset captures everyday sounds from urban environments, including noises like traffic, human activities, and construction. It is commonly used for acoustic scene classification and environmental sound recognition.

**ESC-50**: ESC-50 is a well-known dataset for environmental sound classification, containing 50 categories of everyday sounds such as animal noises, natural elements, and human activities, making it suitable for multi-class classification challenges.

**UrbanSound8K (USD8K):** USD8K is a curated collection of urban sounds divided into ten classes, including sirens, street music, and car horns. It is used widely for evaluating models on sound event detection in real-world scenarios.

**GTZAN:** GTZAN is a music genre classification dataset that includes ten music genres such as pop, rock, and jazz. It is a standard benchmark for evaluating music classification models, although it has known data quality issues.

**Medley-solos-DB:** This dataset consists of solo recordings of different musical instruments, making it valuable for studying isolated instrument sounds and training models for music instrument recognition.

**MUSDB18:** MUSDB18 is used primarily for music source separation tasks. It contains full-track recordings of different music styles, providing a mix of vocals, drums, bass, and other instruments, useful for multi-class classification.

**DCASE Task 4:** Part of the DCASE challenge, this dataset focuses on domestic sound scene and event classification. It includes various audio clips recorded in home environments, often used for anomaly detection and sound event classification.

**Vocal Sounds (VS):** This dataset includes various vocal sounds such as singing, speech, and vocal effects, providing rich data for studying voice classification and enhancing models for vocal audio recognition tasks.

## A.6 ALGORITHM

Algorithm 1 algorithmically illustrated Synthio.

---

**Algorithm 1** Synthio Framework for Audio Classification Augmentation

---

**Require:** Small human-annotated dataset $\mathcal{D}_{\text{small}}$;
    Noisy audio-caption paired dataset $\mathcal{D}_{\text{a-c}}$;
    Number of generations per instance $j$;
    Similarity threshold $p\%$;
    Maximum self-reflection iterations $i_{\max}$.

**## Initial Training of T2A Model**
Train T2A model $\mathcal{T}^\theta$ on $\mathcal{D}_{\text{a-c}}$.

**## Construction of Preference Dataset $\mathcal{D}_{\text{pref}}$**
**for** each audio instance $d_k$ in $\mathcal{D}_{\text{small}}$ **do**
    Create caption $c_k$ = "Sound of a $label_k$".
    **for** $l$ = 1 to $j$ **do**
        Generate audio $\tilde{a}_{k,l} = \mathcal{T}^\theta(c_k)$ starting from random noise.
        Pair $(\tilde{a}_{k,l}, a_k)$ where $a_k$ is the ground-truth audio.
        Add pair to $\mathcal{D}_{\text{pref}}$ with $\tilde{a}_{k,l}$ as loser and $a_k$ as winner.
    **end for**
**end for**

**## Preference Optimization Using DPO**
Fine-tune $\mathcal{T}^\theta$ on $\mathcal{D}_{\text{pref}}$ using DPO methodology.

**## Generating Diverse Prompts with MixCap**
Use audio captioning model to generate captions for all $a_k$ in $\mathcal{D}_{\text{small}}$.
Prompt LLM to extract acoustic components (backgrounds, events, their attributes and relations) from captions.

**for** each label $label_k$ in $\mathcal{D}_{\text{small}}$ **do**
    Using extracted acoustic elments, prompt LLM to generate $n$ diverse captions $\{c_{k,1}, c_{k,2}, \ldots, c_{k,n}\}$.
**end for**

**## Generation of Synthetic Data $\mathcal{D}_{\text{syn}}$**
Initialize $\mathcal{D}_{\text{syn}}^{\text{acc}} \leftarrow \varnothing$, $\mathcal{D}_{\text{syn}}^{\text{rej}} \leftarrow \varnothing$.
**for** each caption $c_{k,m}$ **do**
    Generate audio $\tilde{a}_{k,m} = \mathcal{T}^\theta(c_{k,m})$.
    Evaluate similarity $s_{k,m} = \text{CLAP}(\tilde{a}_{k,m}, label_k)$.
    **if** $s_{k,m} \geq p\%$ **then**
        Add $(\tilde{a}_{k,m}, label_k)$ to $\mathcal{D}_{\text{syn}}^{\text{acc}}$.
    **else**
        Add $(c_{k,m}, label_k)$ to $\mathcal{D}_{\text{syn}}^{\text{rej}}$.
    **end if**
**end for**

**## Self-Reflection and Caption Revision**
Set iteration count $i \leftarrow 0$.
**while** $\mathcal{D}_{\text{syn}}^{\text{rej}} \neq \varnothing$ and $i < i_{\max}$ **do**
    $i \leftarrow i + 1$.
    **for** each rejected caption $c_{k,m}$ in $\mathcal{D}_{\text{syn}}^{\text{rej}}$ **do**
        Provide LLM with $c_{k,m}$ and insights from $\mathcal{D}_{\text{syn}}^{\text{acc}}$.
        Obtain revised caption $c'_{k,m}$.
        Generate audio $\tilde{a}'_{k,m} = \mathcal{T}^\theta(c'_{k,m})$.
        Evaluate similarity $s'_{k,m} = \text{CLAP}(\tilde{a}'_{k,m}, label_k)$.
        **if** $s'_{k,m} \geq p\%$ **then**
            Add $(\tilde{a}'_{k,m}, label_k)$ to $\mathcal{D}_{\text{syn}}^{\text{acc}}$.
            Remove $c_{k,m}$ from $\mathcal{D}_{\text{syn}}^{\text{rej}}$.
        **else**
            Update $c_{k,m} \leftarrow c'_{k,m}$ in $\mathcal{D}_{\text{syn}}^{\text{rej}}$.
        **end if**
    **end for**
**end while**

**## Final Training Dataset and Classification Model**
Combine $\mathcal{D}_{\text{syn}}$ with ground-truth data $\mathcal{D}_{\text{small}}$ to form $\mathcal{D}_{\text{train}}$.
Train audio classification model on $\mathcal{D}_{\text{train}}$.

---

