# OpenReview forum: "Synthio: Augmenting Small-Scale Audio Classification Datasets with Synthetic Data"
_ICLR.cc/2025/Conference — ICLR 2025 Poster_

### Official Review · Reviewer_Q6SZ · 2024-11-04

**Soundness:** 3
**Presentation:** 3
**Contribution:** 2
**Rating:** 6
**Confidence:** 4

**Summary:**

This paper proposed a pipeline to augment small-scale audio classification datasets with synthetic data and shows promising improvements.

**Strengths:**

1. The performance improvement when the data is in small scale (e.g., 50, 100, 200, 500) is pretty significant.
2. The experimental setup is pretty complete and extensive.

**Weaknesses:**

1. The performance improvement with the proposed synthetic pipeline is limited. Despite the proposed augmentation showing quite significant improvement on the curated small dataset with a few hundred samples, it only shows marginal improvement on the full-scale dataset (Table 7).
2. The proposed method is not practical. As we usually have data that is much larger than a few hundred samples (in this case, the proposed method does not work well), the traditionality of the proposed method is questionable. Also the proposed pipeline is much complex comparing with traitional augmentation method such as audiomentations, increasing the cost for model training.
3. Heavy reliance on pretrained CLAP model. This reliance may bias the synthetic data as CLAP model.
4. It is unclear what CLAP threshold is used in filtering and self-reflection. This is usually a tricky operation as the optimal threshold can vary with different types of sound. More explanations are welcomed.

**Questions:**

The result in Section 6.5 and Figure 5 is not convincing enough:
1. Line 516 “ We selected categories with the lowest frequency in the datasets”:
The categories in the ESC-50 dataset have the same number of samples; how do you decide which category has a lower frequency?
2. Figure 5 shows only four classes to demonstrate the improvement of long-tailed categories. This is not convincing enough as those selected classes may represent only partial information.

Table 2: Can you please explain what is “compositional diversity” and how it correlates with the CLAP score?
Line 214: DSmall -> D_{small}
Line 226 “no scalable methods”: Is it possible to estimate the loudness with the signal processing method?
Line 242 “fostering more diverse augmentation”: Any evidence for this claim?
Please double check the equation 8, 9, and 10 as the “x_0 : T” notation looks a bit confusing to me.
Line 318 “The CLAP model is pre-trained on Da-c and fine-tuned on Dsmall to adapt to the target datase”: Do you mean finetuning CLAP model on each of the dataset you used (ESC50, GTZAN, etc)? If so, how do you perform finetuing on such a small dataset, such as when n=50? Doesn’t that can cause overfiting?
Line 413: remove duplicated “the”
What dataset is utilized for the experiment in Table 5? Please specify.
Line 503: What are the possible reasons why SpecAug does not scale well?

Please consider rewording the following sentence that is a bit confusing to read:
- “Abusing notation, we will also x0 as random variables for language”
- “We elaborate on these next”

---

> ### Author Response · Authors · 2024-11-18
> **Response to Official Review of Submission3277 by Reviewer Q6SZ (1/4)**
>
> We thank you for your thorough review and constructive feedback. We have tried to address each of your concerns point by point.
>
> ### Weaknesses
> > Weakness 1
>
> **Ans:** Thank You for bringing up the point! We completely agree with your observation. We would like to emphasize that similar trends have been observed with synthetic data in vision [1,2,3,4] and NLP [5,6,14]. Most synthetic data methods show higher absolute gains in low-resource regimes, and the absolute gains diminish as the number of gold samples increases. Additionally, to support our experiments, we would like to highlight the results from a well known prior work on synthetic data augmentation on the large-scale imagenet: i) with synthetic data augmentation on the large-scale ImageNet [4], authors only show > 1% improvement. We explain our hypothesis behind why this occurs next.
>
> From a scientific perspective, this phenomenon is consistent with findings in prior literature showing that gold data typically captures greater diversity and richness than synthetic data. Gold data inherently reflects the nuanced variations and complex distributions found in real-world scenarios, including subtle contextual and environmental variations that are difficult to replicate synthetically. In contrast, synthetic data, while valuable for filling gaps, often lacks this level of granularity and diversity, especially in capturing long-tail or edge-case instances. As more gold data is added, the need for synthetic data to supplement the dataset's diversity diminishes, as the model increasingly learns from the broader spectrum of examples present in the gold data. This aligns with the observed trend of diminishing returns for synthetic data as the dataset size increases, especially beyond very low-resource settings.
>
>
> > Weakness 2
>
> **Ans:** ``About the practicality of the proposed approach:``
>
> Thank you for your question! We completely agree that usual academic datasets are much larger than a few hundred samples. However, we would like to clarify and address these points:
> - While academic datasets are often large, there are numerous real-world scenarios where labeled data is scarce and expensive to collect or annotate. For instance, in niche applications or highly specific tasks, it is challenging to gather sufficient labeled audio samples. Previous research, such as [3], highlights how synthetic data has proven to be valuable in addressing similar challenges in vision tasks. Synthio demonstrates the potential to enhance audio classification performance in such low-resource scenarios by leveraging just a few annotated audio samples.
>
> - Our work aligns with motivations from few-shot classification and self-supervised learning (SSL), where the focus is on simulating low-resource labeled settings to evaluate algorithmic effectiveness. Even in domains with abundant data, researchers have explored small-scale splits to develop methods that generalize to low-resource scenarios. Similarly, Synthio targets such settings by simulating downsampled versions of benchmark audio datasets, both to showcase its utility in limited labeled data environments and to ensure reproducibility.
>
> - Supporting the earlier point, our experimental design parallels commonly used paradigms in SSL for speech, where researchers have introduced low-resource splits (e.g., 1-hour and 10-hour labeled subsets) even when larger datasets like 960-hour Librispeech are available. Synthio adopts a similar methodology, targeting small-scale labeled data to demonstrate its effectiveness as a synthetic data solution. This approach is crucial for establishing a foundation for synthetic data research in audio classification.
> While Synthio introduces additional computational complexity compared to traditional augmentation methods like audiomentations, the performance benefits in low-resource scenarios outweigh these costs. Similar to SSL approaches, the initial investment in computational resources enables significant performance improvements in cases where labeled data is scarce, making it a valuable tool for practical applications in data-scarce environments.
>
>
> ``About the complexity of the pipeline:``
>
> Thank you for your question. We completely agree with your observation and would like to emphasize the following points to address your concerns:
>
> - A large body of prior research on synthetic data generation in vision [1,2,3,4], NLP [5,6,14], and speech [7,8] follows a similar regime.  These methods utilize powerful generative models trained on large, weakly-labeled corpora to generate synthetic data, which is then combined with the original dataset. Synthio builds on this established paradigm, adapting it effectively for the challenging audio classification tasks. It addresses the gap in prior literature and demonstrates the applicability of synthetic data techniques beyond more established domains like vision and NLP.
>
> (continued in next part..)

---

> ### Author Response · Authors · 2024-11-18
> **Response to Official Review of Submission3277 by Reviewer Q6SZ (2/4)**
>
> (cont..)
>
> - We acknowledge that leveraging synthetic data requires additional resources (data and compute), both for generating the synthetic samples and for fine-tuning the classification model. However, the performance improvements achieved—particularly in low-resource scenarios—demonstrate the value of this trade-off. The significant gains in classification accuracy and robustness outweigh the added computational cost, which is a known and accepted limitation of synthetic data approaches. This is analogous to Self-Supervised Learning (a commonly used method with similar objectives of enhancing performance in small-scale labeled data settings), which also incurs its own limitations, such as requiring additional training on unlabeled data and relying on complex, often uninterpretable algorithms or compute-intensive techniques like large batch sizes.
>
> - To further ensure a fair evaluation, we would like to highlight specific synthetic data baselines in Table 1—Retrieval, Vanilla Syn. Aug., and Vanilla Syn. Aug. + LLM Caps—that also follow the paradigm of expanding small-scale datasets with synthetic data. As mentioned in line 365, these baselines are inspired by prior work on synthetic data for audio classification. Synthio significantly outperforms these baselines, further demonstrating its effectiveness and the impact of its novel contributions. We hope this addresses your concern and provides clarity on the context and advantages of our approach.
>
> > Weakness 3 & 4
>
> **Ans:** Thank you for raising this important point! We appreciate the opportunity to clarify. For CLAP filtering, we use a fixed threshold of $p = 0.85$ across all datasets and experimental settings. This detail has also been included in the revised version of our paper. To address your question, we provide additional experiments and explanations below:
>
> - We have added an ablation study in Appendix A.5, Table 11, where we evaluate Synthio without the CLAP filtering stage (referred to as Synthio w/o CLAP). A snapshot of the results is provided below:
>
> | n   | Method           | ESC-50 | USD8K | GTZAN | TUT   | VS    |
> |-----|------------------|--------|-------|-------|-------|-------|
> | 50  | Synthio          | 49.50  | 76.12 | 68.20 | 43.84 | 80.67 |
> |     | Synthio w/o CLAP | 47.25  | 74.34 | 66.35 | 40.28 | 77.29 |
> | 100 | Synthio           | 83.35  | 85.00 | 71.20 | 71.23 | 86.70 |
> |     | Synthio w/o CLAP | 82.55  | 84.64 | 69.30 | 70.41 | 84.93 |
> | 200 | Synthio           | 86.10  | 82.81 | 82.05 | 56.83 | 87.52 |
> |     | Synthio w/o CLAP | 85.25  | 79.94 | 80.54 | 55.22 | 86.31 |
> | 500 | Synthio           | 92.10  | 89.18 | 82.25 | 67.81 | 91.42 |
> |     | Synthio w/o CLAP | 90.25  | 88.42 | 89.70 | 65.42 | 89.67 |
>
> - As the results indicate, removing the CLAP filtering stage does not lead to a significant drop in performance. The reason for this is explained in the next point.
>
> - Our T2A model uses the same CLAP text encoder for generating audio. Consequently, most generated audios are already highly aligned with the intended category label. However, the purpose of CLAP filtering is to safeguard against cases where the LLM hallucinates and generates a caption that deviates significantly from the intended label. In such cases, CLAP filtering ensures that audios generated from hallucinated captions are discarded, preventing them from negatively impacting the learning process.
>
> - To further address your concerns, we conducted experiments with lower CLAP thresholds ($p = 0.5$ and $p = 0.3$) on three small-scale settings and four datasets. The results are summarized below:
>
> | n   | p    | ESC-50 | USD8K | GTZAN | TUT   | VS    |
> |-----|------|--------|-------|-------|-------|-------|
> | 50  | 0.85 | 49.50  | 76.12 | 68.20 | 43.84 | 80.67 |
> |     | 0.3  | 47.10  | 74.14 | 67.50 | 41.17 | 79.32 |
> |     | 0.5  | 48.25  | 75.39 | 67.75 | 41.93 | 79.48 |
> | 100 | 0.85 | 83.35  | 85.00 | 71.20 | 71.23 | 86.70 |
> |     | 0.3  | 82.55  | 84.64 | 69.30 | 70.41 | 84.93 |
> |     | 0.5  | 82.70  | 84.73 | 70.25 | 70.86 | 85.22 |
> | 200 | 0.85 | 86.10  | 82.81 | 82.05 | 56.83 | 87.52 |
> |     | 0.3  | 85.25  | 79.94 | 80.55 | 55.22 | 86.31 |
> |     | 0.5  | 85.70  | 80.30 | 81.30 | 56.19 | 87.11 |
> | 500 | 0.85 | 92.10  | 89.18 | 82.25 | 67.81 | 91.42 |
> |     | 0.3  | 90.25  | 88.42 | 80.70 | 65.42 | 89.67 |
> |     | 0.5  | 91.65  | 89.07 | 81.05 | 66.35 | 90.02 |
>
> - As observed, Synthio's performance remains robust even with varying thresholds, showing minimal degradation.
>
> - We observed that CLAP filtering led to a maximum reduction of ~10% of the generated samples. This further supports our earlier point that the filtering step primarily serves as a safeguard against rare hallucination cases rather than a critical factor for overall performance.
>
> Finally, all the above points and additional details have been included in Appendix A.4.4 of the revised paper for further reference.
>
> (continued in next part)

---

> ### Author Response · Authors · 2024-11-18
> **Response to Official Review of Submission3277 by Reviewer Q6SZ (3/4)**
>
> ### Questions
> > The result in Section 6.5 and Figure 5 ..:
>
> **Ans:** We apologize for the confusion and are grateful for pointing this out! The selection of categories with the lowest frequency in Section 6.5 only holds for NSynth and not ESC-50. To answer both your questions, we have edited this in the revised version of our paper with 8 categories of NSynth and removed ESC-50.
>
> > Table 2: Can you please explain what is “compositional diversity” and how it correlates with the CLAP score?
>
> **Ans:** Thank you for the question. Compositional diversity refers to the variety and complexity of the individual acoustic events within an audio sample, including their sequence, attributes, and other explicit features that collectively define the structure and richness of the audio.
>
> Compositionality in audio-language models, including CLAP, has been explored in prior work [1]. Specifically, the CLAP score (or semantic similarity) between two audios tends to be low when the audios differ compositionally. Conversely, audios that share similar compositional structures yield higher similarity scores. In this context, we demonstrate that our generated augmentations exhibit significant compositional diversity compared to the original data, as evidenced by their lower CLAP scores, indicating their effectiveness in enriching the dataset with varied and non-redundant augmentations.
>
> > Line 226 “no scalable methods”: Is it possible to estimate the loudness with the signal processing method?
>
> **Ans:** Thank You for the question. Yes it is possible. However, when we wrote “there are no scalable methods for extracting specific attributes for each label,” we meant two things: 1) if we extract such attributes for every instance in the dataset, there is no scalable method to generalize it to a few natural language words across all instances (not all audios with the  may be loud. 2) “Loud” for one label may not be loud for another. This makes it difficult to generalize such words. T2A models cannot still take intensity of loudness (or any other acoustic characteristics as input). Our proposed method overcomes these problems with DPO. DPO alignment makes sure the acoustic characteristics are aligned and “sound” similar to the downstream dataset.
>
> Finally, our second point “T2A models struggle with accurately following fine grained prompt details” emphasizes that even if we have such natural language words for our text prompts, T2A models will not be able to follow them.
>
> > Line 242 “fostering more diverse augmentation”:
>
> **Ans:** Thank You for the question. Yes, our comparison between Synthio and Synthio w/ ERM in Table 2 shows this! We highlight the results below:
>
> | n | Method | ESC-50 | USD8K | GTZAN | Medley | TUT | NSynth | VS | MSDB | DCASE | FSD50K|
> |-|-|-|-|-|--|--|-|-|-|-|-|
> | 50| Synthio (ours) | 49.50 | 76.12 | 68.20 | 60.58 | 43.84 | 40.83 | 80.67 | 60.15 | 17.23 |  15.41 |
> | | w/ ERM | 41.30 | 69.80 | 61.70 | 56.60 | 42.00 | 38.62 | 79.75 | 57.75 | 13.28 | 14.17
> | 100 | Synthio (ours) | 83.35 | 85.00 | 71.20 | 71.23 | 52.42 | 44.92 | 86.70 | 68.80 | 19.38 | 17.33 |
> | | w/ ERM | 73.20 | 81.81 | 67.25 | 66.57 | 51.11 | 43.74 | 84.73 | 68.00 | 17.21 | 15.19 |
> | 200|Synthio (ours) | 86.10 | 82.81 | 82.05 | 79.40 | 56.83 | 57.10 | 87.52 | 80.40 | 32.81 | 21.60 |
> | | w/ ERM | 85.35 | 79.82 | 80.20 | 74.43 | 55.76 | 56.15 | 86.92 | 74.40 | 29.81 | 19.26 |
> | 500 |Synthio (ours) | 92.10 | 89.18 | 82.25 | 78.62 | 67.81 | 65.40 | 91.42 | 74.70 | 39.24 | 24.93 |
> | | w/ ERM | 91.20 | 88.25 | 79.15 | 77.38 | 65.80 | 64.27 | 88.74 | 74.20 | 38.03 | 23.72 |
>
> Specifically, we claim that standard fine-tuning (w/ ERM) leads the model to overfit to the downstream dataset. On the other hand, DPO-based alignment only aligns generations  or makes the sounds in the downstream dataset sound similar in the synthetic data by promoting similar acoustic characteristics. Thus, as mentioned previously, a higher CLAP score for standard fine-tuning (w/ ERM) shows that audios have low diversity. On the other hand, a lower CLAP score for DPO generate more divers audios
>
> > Please double check the equation 8, 9, and 10 as the “x_0 : T” notation looks a bit confusing to me.
>
> Thank You so much for the comment. For better clarity, we have now moved T to the subscript and made it to x_{0:T} in the revised version of our paper. $T$ stands for the diffusion timesteps from 0 to T.
>
> > Line 318 “The CLAP model is pre-trained on Da-c and fine-tuned on Dsmall to adapt to the target datase”: Do you mean finetuning CLAP model on each of the dataset you used (ESC50, GTZAN, etc)? If so, how do you perform finetuing on such a small dataset, such as when n=50? Doesn’t that can cause overfiting?
>
> **Ans:** Thank You for the comment! We missed adding the detail that we only fine-tune the last layer of CLAP and keep all other layers frozen. We have added this detail in the final version of our paper (lines 317 - 318). Our code in the Appendix reflects the same.

---

> ### Author Response · Authors · 2024-11-18
> **Response to Official Review of Submission3277 by Reviewer Q6SZ (4/4)**
>
> > Line 413: remove duplicated “the”
>
> Thank You for the suggestion. We have removed this in the revised version of our paper.
>
> > What dataset is utilized for the experiment in Table 5? Please specify.
>
> **Ans:** Thank You for the question! We employed a randomly selected subset of the AudioCaps dataset for evaluation. We have uploaded the dataset in the supplementary material and will opensource all data splits upon paper acceptance.
>
> > Line 503: What are the possible reasons why SpecAug does not scale well?
>
> **Ans:** SpecAugment applies basic transformations (like time-warping, frequency masking, etc.) to the original audio, which don’t introduce new semantic or acoustic elements (or compositional diversity). These transformations are limited in diversity because they only alter the original audio in simple ways, so they don’t significantly expand the dataset’s variability. As a result, scaling up SpecAugment doesn’t add meaningful new information to the model and may even lead to overfitting on repetitive patterns.
>
> On the other hand, synthetic data generated by T2A models like Synthio introduces genuinely new samples that capture different variations of the target classes. Each generated sample can represent unique combinations of acoustic elements (e.g., background noise, timbre variations, event co-occurrences) and potentially fill in gaps in the dataset's distribution.
>
> Finally, we would like to emphasize that our experiments show how SpecAug-like transformations are effective on large-scale datasets (due to the availability of already diverse samples in the corpus) where basic transformations might work fine. Synthetic data (especially with Synthio) proves to be a more effective solution in low data (small-scale) regimes.
>
> ## References
>
> [1] IS SYNTHETIC DATA FROM GENERATIVE MODELS READY FOR IMAGE RECOGNITION? (https://openreview.net/forum?id=nUmCcZ5RKF – ICLR 2023)
>
> [2] Expanding Small-Scale Datasets with Guided Imagination (https://openreview.net/forum?id=82HeVCqsfh – NeurIPS 2023)
>
> [3] Effective Data Augmentation With Diffusion Models (https://openreview.net/forum?id=ZWzUA9zeAg – ICLR 2024)
>
> [4] Synthetic Data from Diffusion Models Improves ImageNet Classification (https://openreview.net/forum?id=DlRsoxjyPm – TMLR 2023)
>
> [5] ABEX: Data Augmentation for Low-Resource NLU via Expanding Abstract Descriptions (https://aclanthology.org/2024.acl-long.43/ – ACL 2024)
>
> [6] PromDA: Prompt-based Data Augmentation for Low-Resource NLU Tasks (https://aclanthology.org/2022.acl-long.292/ – ACL 2023)
>
> [7] Speech Self-Supervised Learning Using Diffusion Model Synthetic Data (https://openreview.net/forum?id=ecnpYYHjt9 – ICML 2024)
>
> [8] Generating Data with Text-to-Speech and Large-Language Models for Conversational Speech Recognition (https://www.isca-archive.org/syndata4genai_2024/cornell24_syndata4genai.pdf – InterSpeech 2024)
>
> [9] Can Synthetic Audio From Generative Foundation Models Assist Audio Recognition and Speech Modeling? (https://arxiv.org/abs/2406.08800)
>
> [10] Synthetic training set generation using text-to-audio models for environmental sound classification (https://arxiv.org/abs/2403.17864)
>
> [11] Libri-Light: A Benchmark for ASR with Limited or No Supervision (https://arxiv.org/abs/1912.07875)
>
> [12] STABLE DISTILLATION: REGULARIZING CONTINUED PRE-TRAINING FOR LOW-RESOURCE AUTOMATIC SPEECH RECOGNITION (https://arxiv.org/pdf/2312.12783v1)
>
> [13] Domain Gap Embeddings for Generative Dataset Augmentation (https://openaccess.thecvf.com/content/CVPR2024/papers/Wang_Domain_Gap_Embeddings_for_Generative_Dataset_Augmentation_CVPR_2024_paper.pdf – CVPR 2024)
>
> [14] ABEX: Data Augmentation for Low-Resource NLU via Expanding Abstract Descriptions (https://aclanthology.org/2024.acl-long.43/ – ACL 2023)
>
> [15] Deep Long-Tailed Learning: A Survey (https://arxiv.org/pdf/2110.04596 – TAPMI)

---

> ### Author Response · Authors · 2024-11-22
> **Request to review the rebuttal**
>
> Dear reviewer Q6SZ,
>
> Thank you for taking the time to review our paper. We have addressed your concerns in our submitted response and provided a revised version of the paper. As the rebuttal period is nearing its conclusion, we kindly request you to review our rebuttal and share any additional comments or concerns you may have. Thank you once again for your valuable feedback!
>
> Best,
> Authors of Submission3277

---

> ### Author Response · Authors · 2024-11-23
> **Request to review the rebuttal**
>
> Dear reviewer Q6SZ,
>
> Thank you for taking the time to review our paper. We have addressed your concerns in our submitted response and provided a revised version of the paper. As the rebuttal period is nearing its conclusion, we kindly request you to review our rebuttal and share any additional comments or concerns you may have. Thank you once again for your valuable feedback and we would be happy to answer additional questions!
>
> Best,
> Authors of Submission3277

---

> > ### Comment · Reviewer_Q6SZ · 2024-11-24
> > **Thanks**
> >
> > Thanks for addressing my review comments. I am happy to raise my score.

---

> > > ### Author Response · Authors · 2024-11-25
> > > **Thank You**
> > >
> > > We are glad to know that we could resolve your concerns. We thank you for your invaluable feedback which will help us improve the quality of our paper. We really appreciate the score increase.
> > >
> > > Best,
> > > Authors of Submission3277

---

### Official Review · Reviewer_8CiF · 2024-11-07

**Soundness:** 1
**Presentation:** 3
**Contribution:** 2
**Rating:** 5
**Confidence:** 5

**Summary:**

The paper proposes a straightforward method: Given small scale audio classification task - use an existing audio generation model in this paper, a text-to-audio generator, to generate a lot of audio files similar to the smaller dataset and augment it with a factor of 1x to 5x samples to fine-tune an existing audio classification. The authors show the method outperforms their proposed baseline by (relative) 0.1-39%. The authors say that their contributions are,

 -- it is a novel approach for augmenting the small-scale dataset
--  their ability to generate novel, diverse, consistent data and its performance on tasks such as audio captioning.

In brief, this is a paper on audio classification on small-scale audio. The methodology is once the dataset is created == straightforwardly fine-tune the model.

**Strengths:**

== The paper has extensive experimentation that is far more than what was needed. It would have been much stronger if the authors focussed on a subset of these results with a sound, solid method.

== The paper appears to be well-structured and written despite its flaws.

== The data and code will be open-sourced, reproducible and easy to follow, which is a good point in favour of the paper.

== A well-explained background on DPO, diffusion model, and RLHF. It should have included background for audio classification, zero-shot learning, etc.

== The problem is interesting; exploring synthetic data to improve audio classification and pushing the performance is worth exploring.

**Weaknesses:**

The paper is being submitted to the topmost ML venue; hence, my comments are made accordingly to adhere to the highest quality of the paper. At the outset, the paper appears strong, but there are flaws that need to be addressed / claims that are not correct. Please do not carry out extra experiments to justify any other points and generate a new version of the paper. Please answer these points in 1-2 lines of text at maximum.

1. The authors should drop a novel approach for audio classification as several papers exist that should be referenced. i) Open-Set Tagging Dataset (OST) Mark Cartwright, ii) BLAT Bootstrapping language-audio pertaining based audio set .... iii) Can synthetic audio from generative foundational models assist audio recognition ....iv) Synthetic training set generation using T2A models for environmental sound classification. There exist many more papers for audio generation tasks, which I do not include here, but the authors should have included them. These should be adequately referenced in the related work. If not, it gives a very false impression that the authors are the first ones to do it, including in the abstract.

2. The whole point of audio classification is robustness to unseen conditions and good performance for out-of-distribution data. DPO aligns generated data with the small dataset but fails to capture the variations that are needed in real life, as claimed by the author. The claim made by the authors, "T2A models trained on internet-scale data often generate audio that diverges from the characteristics of
small-scale datasets, resulting in distribution shifts.
These mismatches can include variations in spectral
(e.g., frequency content), perceptual (e.g., pitch, loud-
ness), harmonic, or other acoustic characteristics 2" is correct, but for audio classification, we do want to incorporate all of these variations.


This is the point of training on a large corpus of datasets, such as an audio set, to include all the variations different from the training corpus. If I take very few examples and align a T2A model with DPO to the smaller dataset, it will generate samples in that small dataset distribution, as mentioned in all of the plots. This is shown precisely in Figure 3, and we do not want that. I would rather want a lot of diversity in pitch and spectral flux to make audio classification good in unseen cases rather than have the training data similar to my small corpus.

There is a whole subfield on "out of distribution classification" = how to make models perform well when the classification dataset is not of the same distribution as that of the original training dataset. Whereas here, the authors are trying to do just the opposite and make the training corpus as close to the actual data using DPO, which is counter-intuitive to the field of how to build robust classifiers and get good classification accuracy out-of-distribution examples and data.

3. Audiomentations has a list of 40 possible audio augmentations. The setup with n being 50-500 for ten categories would not be able to generate enough diverse augmentations. How do the authors report the numbers for it and carry it out? With N being at max 5, how would one incorporate it? Are all of the combinations not needed/not needed? how were they chosen?

4. From Table 1, it does not appear that this paper was an academic paper. The comparisons do not include any benchmark reported by any other paper. The baselines are poorly reported and are not compared to any other paper. Even for data augmentation literature, there are papers that compare various techniques.

5. Section 6.4 mentions that the scaling 4x is better than 5x. This appears strange as to why this would happen. The audio captioning section is definitely not needed for this paper and should ideally be dropped.

6. The issue with alignment to the dataset is also the fact that there are categories that often overlap in FSD-50K, ESC-50, etc. What happens in that setup? In that case, the model is forced to generate one kind of audio sample/or prefer one over the other even though both have the same label.

7. Accuracy mentioned in 6.5 is 0%. It is mentioned that the categories had the lowest frequencies in the dataset. This makes the claims in the paper shaky as it appears that the training was not normalized according to the category probabilities or sub-sampled/oversampled to make the training balanced.

**Questions:**

1. How would aligning the T2A model with the small dataset create the diversity of audio as exists in the real world as claimed in the Abstract as DPO would make it close to the samples chosen.

2. Why was random mix augmentation not carried out in the Table 1 ?

3. Was the performance boost mentioned in the abstract 0.1-39% relative or absolute. I could not find that in Table 1. Why were MAP metrics not report as is a standard practice for FSD-50K

4. Why are the characteristics like spectral flux and spectral flatness done globally as opposed to local behavior of NSynth and Urban Sound dataset.

5. Are there any papers that have the experimental setup similar to the Table 1. Please cite a paper that fine-tunes such a small corpus in a similar setup.

6. Why was the whole background on few-shot classification and zero shot audio classification not included. This is very similar to zero shot and few shot classification. The baselines should be included as well. Why should someone generated synthetic examples and not do few shot audio classification. What are the drawbacks of those methods. Why are the baselines not included with any of the previous works on synthetic dataset. n=50  is a few-shot setting, and it would be unfair not to include any literature or mention it anywhere in the paper.

7. Why have the authors gone the T2A route. Taking an audio and writing a caption is an extremely lossy compression which will remove most of the information needed about the audio. what were the advantages of a T2A model that will be helpful for this problem.

8. Why was the scaling factor limited to only 1x-5x scaling. What happens if I do 20x. It seems it might start overfitting ? The results in 6.4 are strange that it works with 4x scaling than 5x scaling. Either the models are not tuned properly or they are overfitting on smaller corpus.

---

> ### Author Response · Authors · 2024-11-18
> **Response to Official Review of Submission3277 by Reviewer 8CiF (1/4)**
>
> We thank you for your thorough review and constructive feedback. We have tried to address each of your concerns point by point. Acknowledging your request, we have tried our best to keep the answers short. However, some responses might be longer and e apologize for that. We have also tried to keep the revisions minimal, but had to revise a few things to address the comments of other reviewers.
>
> ### Weaknesses
> > Weakness 1
>
> **Ans:** Thank you for the suggestion! We appreciate the references and will include them in the revised version of the paper. However, we would like to clarify that our primary focus is on improving the quality of synthetic data generation to enhance audio classification performance in a standard evaluation setup (e.g., fine-tuning a standard encoder).
>
> Additionally, similar trends are observed in related fields. For example, prior work in audio SSL [18] (which shares a similar objective of improving classification performance in low-resource labeled settings) primarily focuses on proposing SSL algorithms, without introducing new classification methods. Our motivation and setup are aligned with synthetic data methods in vision [1,2,3,4], NLP [5,6,14], and speech [7,8], which similarly aim to improve synthetic data generation for downstream tasks.
>
> Part 2:
>
> Thank You for pointing us to the baselines. We have referenced some work in Section 2. However, here are some further points which we would like you to please consider.
>
> - We do cite ``Francesca Ronchini, Luca Comanducci, and Fabio Antonacci. Synthesizing soundscapes: Leveraging text-to-audio models for environmental sound classification. arXiv preprint arXiv:2403.17864, 2024.`` in Section 2, which is one of the works to explore synthetic data for audio classification. We have also employed that as a baseline for comparison in all Tables (named Vanilla Syn. Aug. baseline). iv) seems to be the same paper but the authors have changed the name of the paper)
> - iii) is very similar to iv) but with LLM generated captions. We are aware of this work and have thus compared Vanilla Syn. Aug. + LLM captions in Table 1. We apologize for missing to cite it explicitly and we have cited the paper in the revised version now.
> - While i) and ii) are not exactly related to improving audio classification with synthetic data (and are more related to research in the audio-language space), we have cited all these papers in the revised version of our paper and discussed how they perform synthetic caption generation for training or classification.
>
> Finally, we would like to emphasize that we never claim that we are the first to explore synthetic data for audio classification. We always write “We propose Synthio, a novel …” which implies it is more novel and effective than existing methods. If you have explicit lines you want us to rewrite please let us know and we would be happy to revise!
>
> >  Weakness 3
>
> **Ans:** Thank You for the question. We think there might be some misunderstanding and apologize for any. For Audiomentaions (and all other traditional augmentations) $N$ does not play any role. These augmentations are initialized and applied on-the-fly during training at each epoch according to their original implementation.
>
> > Weakness 4
>
> **Ans:** Thank you for your question. We address your points as follows:
> About Benchmark:
>  - Unlike speech recognition, where standardized small-scale labeled data benchmarks (e.g., LibriLight 1hr, 10hr for LibriSpeech) exist, we could not identify similar benchmarks for small-scale audio classification. If you have recommendations, we would greatly appreciate them.
> - As one of the first works in this space, we opted to downsample commonly used datasets for our experiments. This approach is consistent with extensive prior work in NLP and vision [4,5,6,14], where well-known datasets are downsampled to simulate low-resource settings for synthetic data evaluation.
>
> About Baselines:
>
> - Our baselines and benchmarks are partly inspired by other works on employing synthetic data for audio classification [9,10] and we report quite a number of baselines as compared to them.
> - We acknowledge that there are a wealth of other baselines in speech processing. However, since our focus is on non-speech sounds and music, we did not compare with speech data augmentation methods. This aligns with prior work on research areas like self-supervised learning, audio classification, and audio generation, which also distinguish between speech and non-speech domains. However, we included SpecAug in our comparisons, as it is widely employed in audio classification tasks.
>
> > Weakness  6
>
> **Ans:** Thank You for the question. This is correct and this is why DPO fine-tuning is done per dataset. For the above example of FSD-50K, ESC-50, DPO fine-tuning would be done per downstream dataset and thus our T2A model would adapt separately.

---

> ### Author Response · Authors · 2024-11-18
> **Response to Official Review of Submission3277 by Reviewer 8CiF (2/4)**
>
> > Weakness 7
>
> **Ans:** Thank you for your question. Long-tailed categories naturally exhibit low accuracy [15], and we believe our classification setup reflects this without issues. As noted in lines 352-354, we preserve the original label distribution (even if unbalanced) with stratified sampling, reflecting real-world conditions where some audio categories are harder to collect.
> Additionally, we follow a standard audio classification setup without class balancing to isolate and evaluate the impact of our synthetic data. This approach aligns with prior work in vision [1,2,3,4], NLP [5,6,14], and speech [7,8], which use similar setups to demonstrate the benefits of synthetic data augmentation.
>
> ### Questions
>
> > Question  1 & Weakness 2
>
> **Ans:** Thank you for your question. We would like to clarify that aligning the T2A model with the small-scale dataset through DPO primarily ensures consistency, not diversity. Specifically, DPO alignment helps the generated audios replicate the implicit acoustic characteristics (e.g., spectral, harmonic, and perceptual features) of the downstream dataset, ensuring they "sound similar" to the original data (see the example in the footer of page 5). This step mitigates the domain shift that arises from the generative model being trained on a broader, more varied dataset like AudioSet, where sounds for a particular category may differ significantly from those in the downstream dataset. Consistency is crucial for effective data augmentation, as prior research in vision and NLP has shown that inconsistent augmentations can lead to catastrophic forgetting and degrade model performance [13] (this is a well studied problem).
>
> On the other hand, achieving semantic diversity is essential for ensuring that synthetic data enriches the dataset with new and meaningful variations. This is addressed by our proposed MixCap module, as described in Section 4.2. MixCap generates audio captions that introduce diverse acoustic elements (e.g., different events, compositional structures, temporal relations) to prompt the T2A model for diverse audio generation. By combining DPO for consistency and MixCap for diversity, Synthio ensures that the generated augmentations are both domain-consistent and semantically diverse, addressing the dual challenges of effective synthetic data generation.
>
> > Question 2
>
> **Ans:** Thank You for the question. We did not find a particular paper that proposes random mix augmentation in audio (though we acknowledge some works use it, like AST). We could have carried out this experiment (as it is part of the AST codebase) but we will respect your request to not perform extra experiments. However, we hypothesize that the performance would be similar to most traditional augmentation methods.
>
> > Question 3
>
> **Ans:** Thank You for the question. The performance boost mentioned in the abstract (0.1-39%) is absolute improvements. We have highlighted this in the revision to our paper.
>
> Thank You for the question. Our selection for F1 as an evaluation metric was inspired by other papers that evaluated audio classification performance on F1 [16]. We acknowledge your suggestion and agree that mAP is a more commonly used metric, and we will revise the values in the final version of the paper, if accepted.
>
> **Update on November 24th:** Awaiting response, we have now revised Table 1 of our paper with mAP scores for FSD.
>
>
> > Question 4
>
> **Ans:** Thank you for the question. These characteristics can exhibit significant variability across datasets. Rather than averaging across datasets, we chose to analyze them individually to better capture the unique acoustic behaviors within each domain. For instance, we selected one dataset focused on environmental sounds and another on music to highlight the effect of DPO on these specific acoustic characteristics.
>
> > Question 5
>
> **Ans:** Thank You for the question. As stated earlier, being one of the first works on using synthetic data for improving audio classification performance (a task with no established benchmarks yet), we do not follow any prior papers for our experimental setup. Specifically, we are inspired by 3 factors on our experimental setup:
>
> Recent prior works on synthetic data for audio classification [9,10] employ benchmark datasets. We also employ similar benchmark audio classification datasets (but more in number) and their downsampled versions (as the primary focus of our work is on improving performance on small-scale datasets – where synthetic data proves to be the most effective).
> Extensive prior in synthetic data research in NLP or vision follow a similar experimental setup of downsampling benchmark datasets [2,5,6,14].
> Finally, our experimental design parallels commonly used paradigms in low-resource ASR evaluation, where researchers have introduced low-resource splits of existing benchmark datasets for evaluation (e.g., 1-hour and 10-hour labeled LirbiLight subsets for LibriSpeech evaluation [11], or SwitchBoard [12])

---

> ### Author Response · Authors · 2024-11-18
> **Response to Official Review of Submission3277 by Reviewer 8CiF (3/4)**
>
> > Question 6
>
> **Ans:** Thank you for your thoughtful comment. We would like to emphasize that few-shot, zero-shot, and synthetic data-based approaches address the challenge of limited labeled data in fundamentally different ways to handle low-resource labeled data settings for audio classification (please see lines 120 - 126 in Section 2). Thus, following common practice in most of synthetic data research in vision [1,2,3,13], NLP [5,6,14], speech [9,10] and audio [7,8], we do not directly compare with methodologically different few-shot and zero-shot methods. For instance, SSL methods, which also aim to improve low-resource classification, are not directly compared with synthetic data approaches in existing benchmarks due to their distinct methodologies. Following this precedent, we chose to focus on synthetic data research, which aligns with our primary scope of generating high-quality augmentations to improve audio classification.
>
> We appreciate your suggestion and have included a detailed discussion of few-shot and zero-shot classification in Section 2 to provide additional context for readers.
>
> Disadvantages:
>
> Few-shot and zero-shot audio classification approaches rely heavily on pre-trained models and often require task-specific adaptation, which may struggle with domain shifts or long-tailed categories due to limited labeled data. In contrast, synthetic data directly increases data diversity and addresses distribution gaps by generating semantically diverse examples, overcoming the reliance on pre-trained embeddings and improving robustness in low-resource and diverse settings.
>
> Copying  lines 120 - 126 in Section 2 for readability of reviewer:
> `Few-shot audio classification focuses on training models to classify audio samples with very limited labeled data per class, often leveraging transfer learning or meta-learning approaches. In contrast, zero-shot audio classification enables models to generalize to unseen categories without direct training on those classes, relying on learned representations or external knowledge. Synthetic data research complements these by generating additional labeled data, improving model performance under low-resource settings while addressing data scarcity without directly requiring labeled instances from the target categories.`
>
> However, we have now added this to the background of our final version of the paper.
>
> > Question 7
>
> **Ans:** T2A models enable the generation of audios with compositional diversity. For instance, by using captions centered around a specific label, a T2A model can create diverse audio variations aligned with that label.
> To address the potential drawback of lossy compression when mapping audio to text (loss of nuanced acoustic details), we incorporate the DPO alignment stage. This ensures the T2A model is aligned with the implicit acoustic characteristics of the downstream dataset, preserving crucial information and boosting the effectiveness of the generated synthetic data.
>
> > Question 8 and Weakness 5
>
> **Ans:** Thank you for your question. The observed phenomenon is consistent with prior works in NLP and vision [4,5,6,14], where the effect of synthetic data augmentation plateaus or diminishes due to limited diversity in generated augmentations, leading to overfitting. For instance, [4] (Fig. 6) shows how increasing synthetic augmentations on ImageNet results in plateauing performance, even with a diverse dataset used for fine-tuning the generation model.
>
> While Naive synthetic augmentation plateaus sooner (see Fig. 1 and Table 1), Synthio sustains gains longer due to its improved diversity. The trend does not indicate improper model tuning but highlights the natural limitations of scaling synthetic data.
> We limited scaling to 1x–5x due to computational constraints and prior findings that synthetic data methods often plateau around these levels.
>
> ### References
>
> [1] IS SYNTHETIC DATA FROM GENERATIVE MODELS READY FOR IMAGE RECOGNITION? (https://openreview.net/forum?id=nUmCcZ5RKF – ICLR 2023)
>
> [2] Expanding Small-Scale Datasets with Guided Imagination (https://openreview.net/forum?id=82HeVCqsfh – NeurIPS 2023)
>
> [3] Effective Data Augmentation With Diffusion Models (https://openreview.net/forum?id=ZWzUA9zeAg – ICLR 2024)
>
> [4] Synthetic Data from Diffusion Models Improves ImageNet Classification (https://openreview.net/forum?id=DlRsoxjyPm – TMLR 2023)
>
> [5] ABEX: Data Augmentation for Low-Resource NLU via Expanding Abstract Descriptions (https://aclanthology.org/2024.acl-long.43/ – ACL 2024)
>
> [6] PromDA: Prompt-based Data Augmentation for Low-Resource NLU Tasks (https://aclanthology.org/2022.acl-long.292/ – ACL 2023)
>
> [7] Speech Self-Supervised Learning Using Diffusion Model Synthetic Data (https://openreview.net/forum?id=ecnpYYHjt9 – ICML 2024)
>
> [8] Generating Data with Text-to-Speech and Large-Language Models for Conversational Speech Recognition (https://www.isca-archive.org/syndata4genai_2024/cornell24_syndata4genai.pdf – InterSpeech 2024)

---

> ### Author Response · Authors · 2024-11-18
> **Response to Official Review of Submission3277 by Reviewer 8CiF (4/4)**
>
> (References cont.)
>
> [9] Can Synthetic Audio From Generative Foundation Models Assist Audio Recognition and Speech Modeling? (https://arxiv.org/abs/2406.08800)
>
> [10] Synthetic training set generation using text-to-audio models for environmental sound classification (https://arxiv.org/abs/2403.17864)
>
> [11] Libri-Light: A Benchmark for ASR with Limited or No Supervision (https://arxiv.org/abs/1912.07875)
>
> [12] STABLE DISTILLATION: REGULARIZING CONTINUED PRE-TRAINING FOR LOW-RESOURCE AUTOMATIC SPEECH RECOGNITION (https://arxiv.org/pdf/2312.12783v1)
>
> [13] Domain Gap Embeddings for Generative Dataset Augmentation (https://openaccess.thecvf.com/content/CVPR2024/papers/Wang_Domain_Gap_Embeddings_for_Generative_Dataset_Augmentation_CVPR_2024_paper.pdf – CVPR 2024)
>
> [14] ABEX: Data Augmentation for Low-Resource NLU via Expanding Abstract Descriptions (https://aclanthology.org/2024.acl-long.43/ – ACL 2023)
>
> [15] Deep Long-Tailed Learning: A Survey (https://arxiv.org/pdf/2110.04596 – TAPMI)
>
> [16] Class-Incremental Learning for Multi-Label Audio Classification - (https://arxiv.org/abs/2401.04447 – ICASSP 2024)
>
> [17] BYOL for Audio: Exploring Pre-Trained General-Purpose Audio Representations (https://ieeexplore.ieee.org/document/9944865 – ICASSP 2024)
>
> [18] Contrastive Learning from Synthetic Audio Doppelgangers (https://arxiv.org/abs/2406.05923v1)

---

> ### Author Response · Authors · 2024-11-22
> **Request to review the rebuttal**
>
> Dear reviewer 8CiF,
>
> Thank you for taking the time to review our paper. We have addressed your concerns in our submitted response and provided a revised version of the paper. As the rebuttal period is nearing its conclusion, we kindly request you to review our rebuttal and share any additional comments or concerns you may have. Thank you once again for your valuable feedback!
>
> Best,
> Authors of Submission3277

---

> ### Author Response · Authors · 2024-11-23
> **Request to review the rebuttal**
>
> Dear reviewer 8CiF,
>
> Thank you for taking the time to review our paper. We have addressed your concerns in our submitted response and provided a revised version of the paper. As the rebuttal period is nearing its conclusion, we kindly request you to review our rebuttal and share any additional comments or concerns you may have. Thank you once again for your valuable feedback and we would be happy to answer additional questions!
>
> Best,
> Authors of Submission3277

---

> > ### Author Response · Authors · 2024-11-25
> > **[Discussion Period Ending Soon] Request to review the rebuttal**
> >
> > Dear reviewer 8CiF,
> >
> > Thank you for taking the time to review our paper. We have addressed your concerns in our submitted response and provided a revised version of the paper. As the rebuttal period is nearing its conclusion, we kindly request you to review our rebuttal and share any additional comments or concerns you may have. Thank you once again for your valuable feedback and we would be happy to answer additional questions!
> >
> > Best,
> > Authors of Submission3277

---

> > > ### Author Response · Authors · 2024-11-27
> > > **[Discussion Period Ending Soon] Request to review the rebuttal**
> > >
> > > Dear reviewer 8CiF,
> > >
> > > Thank you for taking the time to review our paper. We have addressed your concerns in our submitted response and provided a revised version of the paper. As the rebuttal period is nearing its conclusion, we kindly request you to review our rebuttal and share any additional comments or concerns you may have. Thank you once again for your valuable feedback and we would be happy to answer additional questions!
> > >
> > > Best,
> > > Authors of Submission3277

---

> > > > ### Author Response · Authors · 2024-11-28
> > > > **[Discussion Period Ending Soon] Request to review the rebuttal**
> > > >
> > > > Dear reviewer 8CiF,
> > > >
> > > > Thank you for taking the time to review our paper. We have addressed your concerns in our submitted response and provided a revised version of the paper. As the rebuttal period is nearing its conclusion, we kindly request you to review our rebuttal and share any additional comments or concerns you may have. Thank you once again for your valuable feedback and we would be happy to answer additional questions!
> > > >
> > > > Best,
> > > > Authors of Submission3277

---

### Official Review · Reviewer_dSa3 · 2024-11-09

**Soundness:** 3
**Presentation:** 3
**Contribution:** 3
**Rating:** 8
**Confidence:** 4

**Summary:**

This paper presents Synthio, a synthetic data generation approach for improving audio classification models. Synthio leverages a pre-trained text-to-audio generation model, it first aligns the generations with acoustic preferences (obtained from a small dataset), then it leverages an LLM to synthetically generate captions for the generated audio. To further improve data quality and model performance, Synthio refines the dataset via rejection sampling.
The authors evaluated their approach using different benchmarks while generating trainings sets of different sizes. The authors empirically demonstrated the proposed approach is superior to standard data augmentation techniques and to vanilla data generation.

**Strengths:**

1. Leveraging DPO to better align audio generation with real acoustics is a novel idea.
2. The whole data generation pipeline is an interesting idea, including the MixCap approach.
3. Under fine-tuning at small-scale regimes the proposed approach achieves significant improvement over the baseline methods.

**Weaknesses:**

1. Unfair comparisons to the evaluated baselines (more details below).
2. When increasing the number of generated samples the gap in performance between the proposed approach and no aug. at all is closing, very fast.

**Questions:**

As mentioned before, the proposed approach is interesting and novel (in the context of audio generation). However, I have several concerns about the evaluation process and presented results:
1. The authors compare the proposed approach to several baseline methods including Gold-only (meaning no augmentations at all), and more traditional data augmentation techniques. Although the proposed method reaches significantly better results (especially under a small number of samples in D_test), it uses much more resources, meaning, the T2A model was trained on ~1.5M of audio samples. Hence, models trained with Synthio were introduced to more and diverse set of samples than just D_test, which were greatly influenced by the pre-training data of the T2A model.
2. How many samples did you generate for each of the N samples on D_test in Table 1. This is another potential issue as the models trained with Synthio were also trained on more data (nxN) in comparison to standard augmentation techniques which were trained on N samples only.
3. As I mentioned before, it seems the gap in performance is getting smaller very fast as we increase the size of D_test (especially compared to a model with no augmentations). This means that the proposed approach is beneficial for model fine-tuning under very low resource regimes (less than 500 samples). I advise the authors to put more emphasis on that in the paper and discussions.
4. The authors additionally presented results for audio captioning. What text labels were used to both train and evaluate model performance considering both the proposed method and baselines? I'm asking as in the proposed approach text labels are much reacher, hence I'm not sure the comparison is fair.

I'm willing to increase my score in case I missed something.

---

> ### Author Response · Authors · 2024-11-18
> **Response to Official Review of Submission3277 by Reviewer dSa3 (1/2)**
>
> We thank you for your thorough review and constructive feedback. We have tried to address each of your concerns point by point.
>
> ### Questions
> > Question 1
>
> **Ans:** Thank you for your question. We completely agree with your observation and would like to emphasize the following points to address your concerns:
>
> - A large body of prior research on synthetic data generation in vision [1,2,3,4], NLP [5,6,14], and speech [7,8] follows a similar regime. These methods utilize powerful generative models trained on large, weakly-labeled corpora to generate synthetic data, which is then combined with the original dataset. Synthio builds on this established paradigm, adapting it effectively for the challenging audio classification tasks. It addresses the gap in prior literature and demonstrates the applicability of synthetic data techniques beyond more established domains like vision and NLP.
>
> - We acknowledge that leveraging synthetic data requires additional resources (data and compute), both for generating the synthetic samples and for fine-tuning the classification model. However, the performance improvements achieved—particularly in low-resource scenarios—demonstrate the value of this trade-off. The significant gains in classification accuracy and robustness outweigh the added computational cost, which is a known and accepted limitation of synthetic data approaches.
> - This is analogous to Self-Supervised Learning (a commonly used method with similar objectives of enhancing performance in small-scale labeled data settings), which also incurs its own limitations, such as requiring additional training on unlabeled data and relying on complex, often uninterpretable algorithms or compute-intensive techniques like large batch sizes.
>
> - To further ensure a fair evaluation, we would like to highlight specific synthetic data baselines in Table 1—*Retrieval*, *Vanilla Syn. Aug.*, and *Vanilla Syn. Aug. + LLM Caps*—that also follow the paradigm of expanding small-scale datasets with synthetic data. As mentioned in line 365, these baselines are inspired by prior work on synthetic data for audio classification. Synthio significantly outperforms these baselines, further demonstrating its effectiveness and the impact of its novel contributions. We hope this addresses your concern and provides clarity on the context and advantages of our approach.
>
>
> > Question 2
>
> **Ans:** Thank You for your question. We are assuming that you meant D_train and not D_test, as we do not add any samples to D_test. Here are the number of synthetic augmentations generated for different values of $n$ for 3 splits and 3 datasets.
>
> | #n | ESC-50 | GTZAN | TUT |
> |----|--------|-------|-----|
> | 50  | 226    | 221   | 193 |
> | 100  | 486    | 481   | 383 |
> | 200  | 961    | 984   | 948 |
>
> As mentioned in lines 343-345, the number varies according to the dataset and depends on the number of instances after CLAP filtering and the point at which the model starts to overfit or plateau. For the latter part of the question, we request you to please consider our response to the previous question
>
> > Question 3
>
> **Ans:** Thank you for your question. We completely agree with your observation. We would like to emphasize that similar trends have been observed with synthetic data in vision [1,2,3,4] and NLP [5,6,14]. Most synthetic data methods show higher absolute gains in low-resource regimes, and the absolute gains diminish as the number of gold samples increases. Additionally, to support our experiments, we would like to highlight the results from a well known prior work on synthetic data augmentation on the large-scale ImageNet: i) with synthetic data augmentation on the large-scale ImageNet [4], authors only show > 1% improvement. We explain our hypothesis behind why this occurs next.
>
> From a scientific perspective, this phenomenon is consistent with findings in prior literature showing that gold data typically captures greater diversity and richness than synthetic data. Gold data inherently reflects the nuanced variations and complex distributions found in real-world scenarios, including subtle contextual and environmental variations that are difficult to replicate synthetically. In contrast, synthetic data, while valuable for filling gaps, often lacks this level of granularity and diversity, especially in capturing long-tail or edge-case instances. As more gold data is added, the need for synthetic data to supplement the dataset's diversity diminishes, as the model increasingly learns from the broader spectrum of examples present in the gold data. This aligns with the observed trend of diminishing returns for synthetic data as the dataset size increases, especially beyond very low-resource settings.
>
> We have added additional discussions in Appendix A.4.1 and A.4.4 in the revised versions of our paper.

---

> > ### Author Response · Authors · 2024-11-18
> > **Response to Official Review of Submission3277 by Reviewer dSa3 (2/2)**
> >
> > > Question 4
> >
> > **Ans:** Thank You for the question. No text labels are used for our audio captioning experiments. As mentioned in our paper, for generating synthetic data for audio captioning we perform the following steps:
> >
> > Retrain our T2A model on a subset of Sound-VECaps that excludes AudioCaps.
> > Remove the audio captioning module from Synthio
> > Extract the acoustic elements from existing captions in $D_{small}$ (the small-scale audio captioning dataset).
> > Followed by this, we follow the exact same process as the original Synthio pipeline. However, we also skip the CLAP filtering stage (as there is no ground-truth label to compare similarity with).
> >
> > For better clarity, we have revised the lines 482-485 in the paper. Thank You for your suggestion in improving the readability of the paper.
> >
> > ### References
> > [1] IS SYNTHETIC DATA FROM GENERATIVE MODELS READY FOR IMAGE RECOGNITION? (https://openreview.net/forum?id=nUmCcZ5RKF – ICLR 2023)
> >
> > [2] Expanding Small-Scale Datasets with Guided Imagination (https://openreview.net/forum?id=82HeVCqsfh – NeurIPS 2023)
> >
> > [3] Effective Data Augmentation With Diffusion Models (https://openreview.net/forum?id=ZWzUA9zeAg – ICLR 2024)
> >
> > [4] Synthetic Data from Diffusion Models Improves ImageNet Classification (https://openreview.net/forum?id=DlRsoxjyPm – TMLR 2023)
> >
> > [5] ABEX: Data Augmentation for Low-Resource NLU via Expanding Abstract Descriptions (https://aclanthology.org/2024.acl-long.43/ – ACL 2024)
> >
> > [6] PromDA: Prompt-based Data Augmentation for Low-Resource NLU Tasks (https://aclanthology.org/2022.acl-long.292/ – ACL 2023)
> >
> > [7] Speech Self-Supervised Learning Using Diffusion Model Synthetic Data (https://openreview.net/forum?id=ecnpYYHjt9 – ICML 2024)
> >
> > [8] Generating Data with Text-to-Speech and Large-Language Models for Conversational Speech Recognition (https://www.isca-archive.org/syndata4genai_2024/cornell24_syndata4genai.pdf – InterSpeech 2024)
> >
> > [9] Can Synthetic Audio From Generative Foundation Models Assist Audio Recognition and Speech Modeling? (https://arxiv.org/abs/2406.08800)
> >
> > [10] Synthetic training set generation using text-to-audio models for environmental sound classification (https://arxiv.org/abs/2403.17864)
> >
> > [11] Libri-Light: A Benchmark for ASR with Limited or No Supervision (https://arxiv.org/abs/1912.07875)
> >
> > [12] STABLE DISTILLATION: REGULARIZING CONTINUED PRE-TRAINING FOR LOW-RESOURCE AUTOMATIC SPEECH RECOGNITION (https://arxiv.org/pdf/2312.12783v1)
> >
> > [13] Domain Gap Embeddings for Generative Dataset Augmentation (https://openaccess.thecvf.com/content/CVPR2024/papers/Wang_Domain_Gap_Embeddings_for_Generative_Dataset_Augmentation_CVPR_2024_paper.pdf – CVPR 2024)
> >
> > [14] ABEX: Data Augmentation for Low-Resource NLU via Expanding Abstract Descriptions (https://aclanthology.org/2024.acl-long.43/ – ACL 2023)
> >
> > [15] Deep Long-Tailed Learning: A Survey (https://arxiv.org/pdf/2110.04596 – TAPMI)

---

> ### Author Response · Authors · 2024-11-22
> **Request to review the rebuttal**
>
> Dear reviewer dSa3,
>
> Thank you for taking the time to review our paper. We have addressed your concerns in our submitted response and provided a revised version of the paper. As the rebuttal period is nearing its conclusion, we kindly request you to review our rebuttal and share any additional comments or concerns you may have. Thank you once again for your valuable feedback!
>
> Best,
> Authors of Submission3277

---

> > ### Comment · Reviewer_dSa3 · 2024-11-23
> > **Official Comment by Reviewer dSa3**
> >
> > I would like to thank the authors for providing additional clarifications and experiments. I better understand the author's claims and the contribution of this paper. Hence I increased my score.
> >
> > I still highly recommend the authors include a discussion section/paragraph about the relevant use cases of the proposed approach, under what conditions will it be effective, and its limitations.

---

> > > ### Author Response · Authors · 2024-11-23
> > > **Thank You!**
> > >
> > > Dear Reviewer dSa3,
> > >
> > > Thank You for your time in reading the rebuttal. We are glad we could address your concerns! As advised by you, we will add update the paper with a discussion section and revise it in the portal!
> > >
> > > Best,
> > > Authors of Submission3277

---

### Official Review · Reviewer_gsoV · 2024-11-11

**Soundness:** 4
**Presentation:** 4
**Contribution:** 3
**Rating:** 8
**Confidence:** 4

**Summary:**

This paper explores the use of synthetic data to enhance small-scale audio classification tasks, addressing two primary challenges in synthetic augmentation: acoustic consistency and compositional diversity. To improve consistency, a preference alignment approach is introduced to better align the domain of synthetic data with the source dataset. For diversity, the authors propose language-guided audio imagination, utilizing large language models to iteratively generate captions that blend existing and novel acoustic components, feeding into a text-to-audio generation model.

**Strengths:**

- The paper effectively identifies two key challenges in synthetic data-based augmentation, providing clear and well-reasoned explanations.
- The proposed techniques directly address these challenges and are demonstrated to be both practical and beneficial according to the experimental results.
- The experimental setup is comprehensive, covering various types of audio events, which strengthens the validity of the findings.

**Weaknesses:**

The work is well-presented and thorough, with no significant weaknesses noted. However, the reviewer has a few suggestions and minor comments, outlined in the Questions section below.

**Questions:**

- Although the paper strives to present all concepts (with background) in a single, self-contained format, it would benefit from careful notation selection. Reusing the same notation for different concepts could be confusing. For example, using the symbol "beta" in different contexts (e.g., in diffusion, RLHF, and DPO) might lead to misunderstandings.
- Minor correction:  In Table 10, please correct "Stable Stable" to "Stable."
- A major concern with the proposed method is balancing human effort with the complexity of the introduced steps. While the pipeline can operate fully automatically, the large number of hyper-parameters may introduce significant uncertainty in practical applications. The reviewer appreciates the current approach, but simplifying the pipeline to reduce complexity while retaining the core concepts could be a valuable direction for future work.
- Additional experiments:
    - While Synthio shows promising performance, it would be interesting to investigate whether it has complementary or joint effects with other augmentation methods, such as random noise addition, pitch shifting, or other common techniques.
    - Further experiments could examine how changes in T2A models influence the effectiveness of the augmentation process. Specifically, it would be valuable to understand how the quality of T2A models affects the overall performance of the augmentation technique.

---

> ### Author Response · Authors · 2024-11-18
> **Response to Official Review of Submission3277 by Reviewer gsoV**
>
> We thank you for your thorough review and constructive feedback. We're grateful for your recommendation to accept our paper. We have tried to address each of your concerns point by point.
>
> ### Questions
>
> > Although the paper strives to present all concepts (with background) in a single, self-contained format, it would benefit from careful notation selection. Reusing the same notation for different concepts could be confusing. For example, using the symbol "beta" in different contexts (e.g., in diffusion, RLHF, and DPO) might lead to misunderstandings.
>
> **Ans:** Thank you for the suggestions in improving our paper. We have made several changes to the background section to resolve the ambiguities in the revised version of our paper. Please let us know if you still find it confusing.
>
> > While Synthio shows promising performance, it would be interesting to investigate whether it has complementary or joint effects with other augmentation methods, such as random noise addition, pitch shifting, or other common techniques.
>
> **Ans:** Thank You for the question. We present additional results for n=50 and n=100 for 4 benchmark datasets from Table 1. Specifically, we perform Synthio with our traditional augmentation baselines Random Noise, Pitch Shifting, SpecAugment and AudioMentations.
>
> | n   | Method             | ESC-50 | USD8K | GTZAN | Medley |
> |-----|--------------------|--------|-------|-------|--------|
> | 50  | Synthio (ours)     | 49.50  | 76.12 | 68.20 | 60.58  |
> |     | w/ Random Noise    | 49.65  | 77.31 | 70.15 | 61.54  |
> |     | w/ Pitch Shift     | 49.80  | 78.52 | 69.50 | 60.29  |
> |     | w/ Spec Aug        | 50.95  | 77.93 | 70.35 | 61.17  |
> |     | w/ Audiomentations | 50.35  | 77.24 | 69.50 | 61.53  |
> | 100 | Synthio (ours)     | 83.35  | 85.00 | 71.20 | 71.23  |
> |     | w/ Random Noise    | 83.85  | 86.59 | 71.60 | 72.35  |
> |     | w/ Pitch Shift     | 83.60  | 86.32 | 72.95 | 72.50  |
> |     | w/ Spec Aug        | 84.25  | 86.17 | 72.75 | 73.05  |
> |     | w/ Audiomentations | 84.10  | 85.95 | 72.85 | 72.87  |
>
>
> As we can see, performance for baselines boosts significantly with augmentations from Synthio and Synthio + Audiomentations proves to be our best performing approach which performs better than only Synthio.
>
> > Further experiments could examine how changes in T2A models influence the effectiveness of the augmentation process. Specifically, it would be valuable to understand how the quality of T2A models affects the overall performance of the augmentation technique.
>
> **Ans:** Thank You for the question. We show results from 2 experiments: 1) Synthio with Stable Audio-based T2A model trained on AudioCaps (instead of Sound-VECaps in our paper). 2) Synthio with Tango-based T2A model trained on Sound-VECaps. While Stable Audio follows a DiT-based architecture, Tango follows a U-NET architecture.
>
> | n   | Method               | ESC-50 | USD8K | GTZAN | Medley | TUT   |
> |-----|----------------------|--------|-------|-------|--------|-------|
> | 50  | Synthio (ours)       | 49.50  | 76.12 | 68.20 | 60.58  | 43.84 |
> |     | Synthio w/ AudioCaps | 29.20  | 60.15 | 50.15 | 49.19  | 38.62 |
> |     | Synthio w/ Tango     | 48.55  | 75.05 | 66.19 | 59.12  | 42.59 |
> | 100 | Synthio (ours)       | 83.35  | 85.00 | 71.20 | 71.23  | 52.42 |
> |     | Synthio w/ AudioCaps | 58.20  | 74.27 | 66.55 | 67.93  | 48.23 |
> |     | Synthio w/ Tango     | 81.50  | 84.13 | 70.95 | 69.97  | 51.47 |
>
>
> As we can see, replacing Sound-VECaps with AudioCaps leads to a large drop in performance. On the other hand, the drop in performance with Tango-based model is not that significant. Thus, we see that a large enough training data for training the T2A model is more important as it makes the T2A model capable of generating more diverse augmentations.
>
> We have also added these new results to A.4.5 and A.4.6 in the revised version of our paper.

---

> > ### Comment · Reviewer_gsoV · 2024-11-18
> > **Reply to rebuttal**
> >
> > The rebuttal addresses all points raised in the review. It would be useful if the additional experiments and analysis were incorporated into the paper appendix.

---

> > > ### Author Response · Authors · 2024-11-18
> > > **Reply to response**
> > >
> > > Thank You for appreciating the contributions of our paper and acknowledging the responses! We have added all the tables to the Appendix of the revised (and updated) version of our paper (A.4.5 and A.4.6).
> > >
> > > Best,
> > > Authors of Submission #3277

---

### Author Response · Authors · 2024-11-21
**General Response to All Reviewers and Request to Review the Rebuttal**

Dear Reviewers,

We thank the reviewers for their insightful and positive feedback! We are encouraged to find that reviewers gsoV and dSa3 find the idea behind Synthio novel, reviewers gsoV, Q6SZ and dSa3 find our experimental setup complete and extensive and all reviewers find our problem of improving audio classification performance in low-resource settings important.

**In our rebuttal, we have addressed concerns of all reviewers point-by-point. We have also uploaded a revised version of our paper reflecting all the changes. We request all reviewers to please go through our rebuttal and let us know if they have more questions. We would be more than happy to respond to any more questions.**

Best,
Authors of Submission3277

---

### Meta-Review · Area_Chair_m6X5 · 2024-12-22

**Metareview:**

> This paper presents Synthio, a synthetic data generation approach for improving audio classification models. Synthio leverages a pre-trained text-to-audio generation model, it first aligns the generations with acoustic preferences (obtained from a small dataset), then it leverages an LLM to synthetically generate captions for the generated audio. To further improve data quality and model performance, Synthio refines the dataset via rejection sampling. The authors evaluated their approach using different benchmarks while generating trainings sets of different sizes. The authors empirically demonstrated the proposed approach is superior to standard data augmentation techniques and to vanilla data generation.

The results of synthetic data generation for small datasets are promising and useful.

The reviewers mostly agree to accept this paper, in particular with 2 strong accepts, I am inclined to accept the paper.

**Additional Comments On Reviewer Discussion:**

Reviewer 8CiF made a good review and constructive comments. The rebuttal from the authors addresses most of it, although reviewer  8CiF did not change their score. I took this into account.

---

### Decision · Program_Chairs · 2025-01-22

Accept (Poster)